# DAG-MoE: From Simple Mixture to Structural Aggregation in Mixture-of-Experts

Jiarui Feng [1 2 †]   Hanqing Zeng [1]   Karish Grover [3]   Ruizhong Qiu [4]   Yinglong Xia [1]   Qiang Zhang [1]   Qifan Wang [1]
Ren Chen [1]   Dongqi Fu [1]   Jiayi Liu [1]   Zhuokai Zhao [1]   Xiangjun Fan [1]   Benyu Zhang [1]   Yixin Chen [2]

## Abstract

Mixture-of-Experts (MoE) models have become a leading approach for decoupling parameter count from computational cost in large language models, yet effectively scaling MoE performance remains a challenge. Prior work shows that fine-grained experts enlarge the space of expert combinations and improve flexibility, but they also impose substantial routing overhead, creating a new scalability bottleneck. In this paper, we explore a complementary axis for scaling—how expert outputs are aggregated. We theoretically show that replacing the standard weighted-summation aggregation with structural aggregation expands the expert-combination space without altering the experts or router, and enables possible multi-step reasoning within a single MoE layer. To this end, we propose DAG-MoE, a sparse MoE framework that employs a lightweight module to automatically learn the optimal aggregation structure among the selected experts. Extensive experiments under standard language modeling settings show that DAG-MoE consistently improves performance in both pretraining and fine-tuning, surpassing traditional MoE baselines.

## 1. Introduction

Mixture-of-Experts (MoE) models (Shazeer et al., 2017; Lepikhin et al., 2021; Fedus et al., 2022) have become the dominant architecture for large-scale foundation models, including Large Language Models (LLMs). Compared with standard dense models, MoE decouples model size from computational cost by splitting a large dense network into multiple smaller experts, with a router dynamically selecting the top-$K$ most relevant experts for each input. This paradigm has been widely adopted in recent open-source LLMs and multimodal models (Liu et al., 2024; Yang et al., 2025; Meta AI, 2025; Muennighoff et al., 2025; Li et al., 2025b).

Despite the success of MoE, how to effectively scale it remains unclear. Prior work has examined several axes: improving the routing mechanism with more advanced algorithms (Zhou et al., 2022; Qiu et al., 2024; Chi et al., 2022; Wang et al., 2024); analyzing the relationship between MoE performance and sparsity (Li et al., 2025a; Tian et al., 2025); and studying the effect of expert granularity (He, 2024; Ludziejewski et al., 2024). The granularity line, in particular, shows that MoE performance can be improved by increasing both the total number of experts and the number of active experts, while shrinking each individual expert. While promising, this strategy substantially raises router-side complexity and creates a new scaling bottleneck, so state-of-the-art MoE systems typically avoid extremely fine-grained configurations. **All of these efforts focus on the experts and the router, but overlook a third critical component: how expert outputs are aggregated.** In standard MoE, once the router selects the top-$K$ experts, the final representation is formed by a weighted sum using the router scores as coefficients. Because weighted summation is permutation-invariant, the output depends only on the multiset of selected experts, with no notion of ordering or interaction among them, which fundamentally constrains the framework's expressiveness.

In this paper, we propose **replacing simple mixing with structural aggregation**. While prior work largely studies how many experts to keep and which ones to pick, we ask a complementary third question: *given the selected experts, how should their outputs be composed?* We organize the aggregation of the top-$K$ experts into a directed acyclic graph (DAG), assigning each expert a distinct structural role within the graph; expert outputs are then aggregated according to the DAG. We show that introducing such structural relationships effectively expands the expert combination

---

[†] Work done during internship at Meta MRS.  [1] Meta MRS [2] Washington University in St. Louis [3] Carnegie Mellon University [4] University of Illinois Urbana-Champaign. Correspondence to: Hanqing Zeng <zengh@meta.com>, Jiarui Feng <jiaruifeng@meta.com>.

space without modifying the experts or the router, and incurs no additional router-side complexity. Furthermore, the DAG enables the approximation of multi-step reasoning within a single MoE layer, which is beneficial for problems with inherent compositional structure such as dynamic programming. Building on this idea, we present DAG-MoE, an MoE block that incorporates a lightweight DAG learning module: given the top-$K$ experts selected by the router, the module discovers an optimal aggregation structure among them and iteratively refines node representations along the resulting DAG. Across both language modeling and downstream tasks, DAG-MoE consistently outperforms standard MoE architectures, demonstrating the effectiveness of structural aggregation over simple mixing. We release the code in https://github.com/JiaruiFeng/DAG-MoE.

**Conflict of Interest Disclosure.** The authors declare no financial conflicts of interest related to this work.

## 2. Preliminaries

**Architecture of MoE.** In this paper, we consider standard transformer-based LLMs. Let $x \in \mathbb{R}^d$ denote the input token embedding, and let $\{E_i(\cdot) \mid i = 1, \ldots, N\}$ be a set of $N$ expert networks, each implemented as an FFN (Feed-Forward Network) with inner hidden size $d_r$. Let $g_i(\cdot)$ be the router's sparse gating function, which selects the top-$K$ experts for the given input token. The output of the MoE layer is:

$$y = \sum_{i=1}^{N} g_i(x) E_i(x), \tag{1}$$

where the gating function is defined as:

$$g_i(x) = \begin{cases} s_i, & s_i \in \text{TopK}(\{s_j\}_{j=1}^N, K), \\ 0, & \text{otherwise}, \end{cases} \quad s_i = \delta(e_i^\top x). \tag{2}$$

Here $\delta$ is the score function, typically implemented as Softmax or Sigmoid, and $e_i \in \mathbb{R}^d$ is a learnable vector associated with the $i$-th expert. Under this gating, only the top-$K$ experts contribute to the output for each token, while the remaining $N - K$ experts are deactivated.

**Granularity of MoE.** Two key hyperparameters of an MoE model are the total number of experts $N$ and the number of active experts $K$. Recent studies show that increasing the *granularity* of experts—defined as $G = d_f/d_r$, where $d_f$ is the hidden size of the dense FFN counterpart and $d_r$ is the hidden size of each expert—can substantially improve MoE performance (He, 2024; Ludziejewski et al., 2024). A fine-grained MoE keeps the total parameter budget fixed (e.g., $d_f$ fixed) while shrinking $d_r$, which permits a larger $N$ and $K$. Higher granularity dramatically enlarges the space of possible expert combinations: for instance,

choosing top-2 out of 8 experts gives only $\binom{8}{2} = 28$ combinations, whereas choosing top-4 out of 16 experts already yields $\binom{16}{4} = 1{,}820$. However, scaling $N$ also expands the router-side parameter count and load-balance complexity, and state-of-the-art MoE systems therefore avoid extremely fine-grained configurations in practice, which motivates us to seek complementary axes for improving MoE.

## 3. DAG-MoE: Harnessing the power of structure in MoE

### 3.1. From simple mixture to structural aggregation

Given the limitations of fine-grained MoE scaling, we explore complementary ways to enhance MoE capacity. We begin by examining the expressiveness of the standard MoE architecture. As shown in Eq. 1, expressiveness is determined by two components: the expert output set $\mathcal{F} = \{E_i(x) \mid i = 1, \ldots, N\}$ and the score set $\mathcal{S} = \{g_i(x) \mid i = 1, \ldots, N\}$. This further breaks down into the per-element capacity of each $E_i(x)$ and $g_i(x)$, and the cardinalities $|\mathcal{F}| = |\mathcal{S}| = N$. Each $g_i(x)$ is a scalar with no learnable parameters of its own, while each $E_i(x)$ is an FFN of identical architecture whose capacity is determined by its parameter count. Since increasing each expert's parameter count raises practical concerns (compute, memory, deployment), we assume throughout this discussion that the total parameter budget for $\mathcal{F}$ is fixed.

Under this constraint, improving expressiveness reduces to scaling $N$, which is precisely the granularity axis discussed in the previous section. Yet even with $\mathcal{F}$ and $\mathcal{S}$ fully determined, the overall expressiveness is still bounded by the functional form of Eq. 1: once the router has chosen the top-$K$ experts, the output is fully determined, since the weighted sum is permutation-invariant in the selected experts. This raises a natural question: *is there a more effective way than summation to combine information from the selected experts?*

To answer this question, we view the experts from a structural perspective and consider the case where the router selects four experts. As shown on the left of Fig. 1, the aggregation in standard MoE can be interpreted as a computational graph in which each expert corresponds to an isolated node with no edges. In this setting, permuting the order of experts does not change the graph, since all nodes share the same structural role. Let $AGG$ denote a generic aggregation function, and consider a tree-structured computation over the four experts. As illustrated in the middle of Fig. 1, the experts are arranged into a depth-2 hierarchy: at the first level, experts 1 and 2 are combined by $AGG$ to produce an intermediate representation, while experts 3 and 4 are simultaneously merged by another instance of $AGG$. At the second level, the two intermediate outputs are

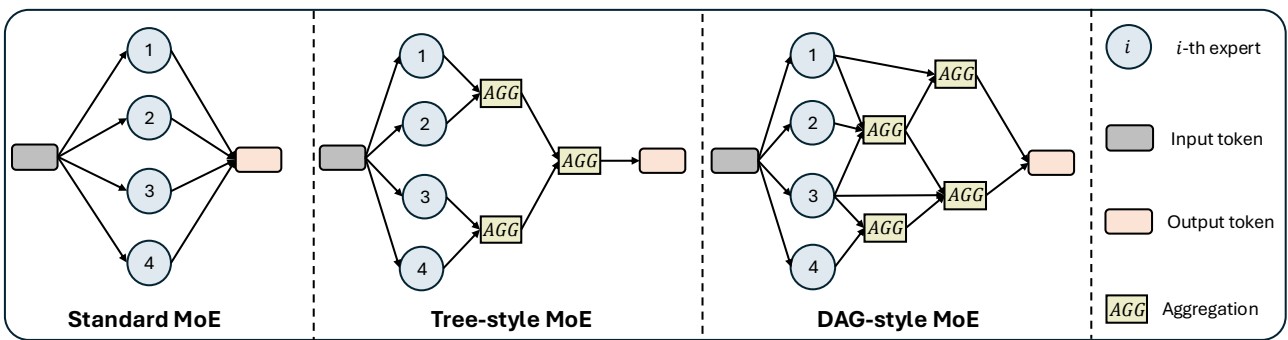

*Figure 1.* Comparison of different mixing structures in MoE.

further combined by yet another instance of $AGG$. In this setup, swapping experts 1 and 3 changes the final output, because the second-level operations now act on different inputs. Hence, experts 1 and 3 occupy distinct structural roles within the expert graph. More generally, the selected experts can be organized into a directed acyclic graph (DAG), with an example shown in the right panel of Fig. 1. Permuting the experts or choosing a different DAG changes the computational graph and yields a different output. Thus, even without modifying the experts or the router, the expressiveness of the MoE architecture can be substantially enhanced through structural composition alone. For a fixed $K$, the number of possible DAGs grows exponentially, offering a rich design space of structural configurations. We refer to this form of structural aggregation as **DAG-style MoE**.

### 3.2. A general formulation of DAG-style MoE

We now formally define and analyze DAG-style MoE. The expert and router configurations remain identical to standard MoE, and we denote the list of top-$K$ experts selected by the router as $\boldsymbol{k} = [k \mid s_k \in \text{TopK}(\{s_j\}_{j=1}^N, K)]$. Let $\mathcal{G}(K)$ be the set of all possible DAGs constructed over $K$ experts, where each DAG is represented as $G = (\mathcal{V}, \mathcal{A})$. Here $\mathcal{V}$ is the set of nodes, and each node $v \in \mathcal{V}$ corresponds to an output representation that is either an initial expert output or an intermediate result produced by the aggregation function $AGG$. Each node is indexed by $v = (l, i)$, where $l$ is its depth and $i$ its index at depth $l$; for example, the initial output of expert 1 corresponds to node $(0, 1)$. The set $\mathcal{A}$ is the adjacency list specifying the connections within the DAG: $\mathcal{A} = \{A_i^l \mid l = 1, \ldots, L; \ i = 1, \ldots, n(l)\}$, where $L$ is the maximum depth and $n(l)$ is the number of nodes at depth $l$. Each $A_i^l$ is the set of predecessor nodes pointing into node $(l, i)$, where $(k, j) \in A_i^l$ means that node $(k, j)$ connects to node $(l, i)$ with $k < l$. For example, the adjacency list of the tree graph in the middle of Fig. 1 is $\mathcal{A} = \{A_1^1 = \{(0, 1), (0, 2)\}, A_2^1 = \{(0, 3), (0, 4)\}, A_1^2 = \{(1, 1), (1, 2)\}\}$.

Let $x_i^l$ denote the learned representation of node $v = (l, i)$. For a given DAG $G \in \mathcal{G}(K)$, the corresponding computa-

tion in DAG-style MoE is formulated as:

$$x_i^0 = g_{\mathbf{k}[i]}(x) E_{\mathbf{k}[i]}(x), \quad i = 1, \ldots, K, \tag{3}$$

$$x_i^l = AGG(\{x_j^k \mid (k, j) \in A_i^l\}), \tag{4}$$
$$i = 1, \ldots, n(l), \ l = 1, \ldots, L-1,$$

$$y = AGG(\{x_j^k \mid (k, j) \in A_1^L\}). \tag{5}$$

Here we assume that the final depth $L$ contains a single node $(L, 1)$ whose connections are specified by $A_1^L$.

### 3.3. Theoretical analysis of DAG-style MoE

With the formulation in place, we present three theoretical results that build on each other to characterize the advantages of DAG-style MoE over standard MoE. Proposition 3.1 first establishes that DAG-style MoE can injectively encode any structure $G \in \mathcal{G}(K)$, which we then use in Theorem 3.2 to show that DAG-style MoE is strictly more expressive than standard MoE. Beyond raw expressiveness, Theorem 3.3 shows that this added capacity translates into a concrete problem-solving advantage: a single DAG-style MoE layer can simulate a full pass of dynamic programming, which a standard MoE layer cannot. Together, these results motivate replacing the permutation-invariant weighted sum with structural aggregation.

**Theoretical expressiveness.** We assume throughout that $AGG$ is a sufficiently powerful injective function over multiset inputs, which can be readily implemented as an MLP combined with sum, min, or max (Zaheer et al., 2017; Xu et al., 2019). Under this assumption, we show that DAG-style MoE is strictly more expressive than standard MoE.

**Proposition 3.1.** *Given a top-$K$ expert list* $\mathbf{k}$, *any DAG-style MoE satisfying Eq. 3–Eq. 5 can injectively encode any* $G \in \mathcal{G}(K)$ *if* $AGG$ *is injective.*

**Theorem 3.2.** *Given a top-$K$ expert list* $\mathbf{k}$, *there exists a DAG-style MoE satisfying Eq. 3–Eq. 5 that is strictly more powerful than the standard MoE in Eq. 1, provided that* $AGG$ *is injective.*

We defer the detailed proofs to Appendix A.1. At a high level, we connect DAG-style MoE to message-passing graph

neural networks (Xu et al., 2019; Gilmer et al., 2017) and leverage results from D-VAE (Zhang et al., 2019) to show that the above formulation can injectively encode any DAG structure and node permutation, whereas standard MoE cannot. This is sufficient to establish that DAG-style MoE is strictly more expressive than standard MoE.

**Benefits for reasoning.** Beyond expressiveness, we now discuss a more practical benefit of DAG-style MoE, using dynamic programming (DP) as a running example. DP is a foundational paradigm for decision-making and combinatorial optimization (we defer its formal definition to Appendix A.2.1). Solving a DP problem amounts to iteratively solving a partially ordered collection of subproblems and aggregating their solutions to produce the final answer. Feng et al. (2023) show that Transformers without a Chain-of-Thought (CoT) mechanism (Wei et al., 2022) cannot effectively solve DP tasks, since a constant-depth model cannot simulate the requisite multi-step subproblem computations. Importantly, many DP procedures induce a DAG over subproblems via their natural partial order, which makes them a direct fit for DAG-style MoE: with sufficient flexibility, a well-trained DAG-style MoE can align its expert DAG with the DP solution graph, where each $AGG$ approximates one subproblem transition and the final readout produces the DP answer. The following theorem formalizes this intuition. Let $G(dp)$ denote the computation DAG of a DP problem and $L(dp)$ its maximum depth.

**Theorem 3.3.** *For any integer $n \in \mathbb{N}$, and any DP problem satisfying Assumptions 4.2–4.5 of Feng et al. (2023) whose total input length $|\boldsymbol{n}|$ is at most $O(K \log n)$, there exists a log-precision Transformer consisting of (i) one multi-head attention layer with $H = O(K \log n)$ heads and (ii) one generic DAG-style MoE block (Equations 3–5) with top-$K$ experts, $L \geq L(dp)$ iterations sharing the same parameters, and adjacency $\mathcal{A}$ specified by $G(dp)$, with hidden dimension $d = O(K \log n)$ and per-iteration parameter count $O(\text{poly}(K))$, such that all parameter values are bounded by $O(\text{poly}(n))$ and the output token is the correct DP answer.*

We defer the detailed proof to Appendix A.3. Briefly, by aligning the DAG-style MoE computation with $G(dp)$, the model can explicitly simulate every intermediate subproblem step for DP instances of input length $O(K \log n)$ (with $K$ active experts), whereas standard MoE can realize only a single aggregation step due to its permutation-invariant summation. Three caveats are worth noting. First, the construction uses a non-constant number of attention heads $H = O(K \log n)$ to gather all input tokens into a single position; this is the cost of collapsing the entire DP execution into a single forward pass. Second, since iterations share parameters, the per-iteration parameter count is independent of $n$, but the overall computation depth $L$ grows with $L(dp)$, which can scale with $|\boldsymbol{n}|$ in the worst case. Third, solv-

ing a full $O(n)$-sized DP instance within a constant-depth forward pass remains out of reach. Nevertheless, a single DAG-style MoE layer can execute multiple reasoning steps, effectively increasing the logical depth of a Transformer layer without significant parameter or compute overhead. Due to space limits, we defer a detailed worked example to Appendix A.2.2.

**Remarks on Theorem 3.3.** Theorem 3.3 is a *capacity* result: it asserts the existence of a parameter setting under which a single DAG-style MoE layer simulates DP solving, much as universal approximation theorems guarantee representational capacity without specifying how gradient descent would find such parameters. The DP construction should therefore be read as a theoretical motivation rather than a prescription: it shows that structural aggregation naturally fits multi-step compositional computation. Correspondingly, we do not claim that the DAG learned by DAG-MoE on natural-language data exactly matches any specific $G(dp)$; we only argue that, by capturing the spirit of hierarchical, ordered aggregation, DAG-style MoE is better positioned to support compositional reasoning than its permutation-invariant counterpart.

### 3.4. DAG-MoE: learning optimal DAG between experts

While organizing experts into a DAG offers clear theoretical advantages, realizing a DAG-style MoE architecture is non-trivial. Since $|\mathcal{G}(K)|$ grows exponentially in $K$, the design space of possible structures quickly becomes intractable. A straightforward workaround is to predefine the structure and tailor the model accordingly. For example, S′MoRE (Zeng et al., 2025) fixes the structure to a tree and employs a hierarchical router that selects experts depth by depth in a top-down manner. However, this mechanism does not generalize to other structures, and different tokens may benefit from different structures. We instead introduce DAG-MoE, a general and practical architecture that learns the DAG structure and performs aggregation accordingly.

**Architecture design.** Given an input token $x$, the router first selects the top-$K$ experts and obtains their output representations, which serve as the initial node set for DAG learning. A DAG learning module is then applied to infer the structure among these expert representations and aggregate them based on the learned connections. This module operates for $L$ iterations; each iteration applies the same module to learn the structure at the current depth, producing a DAG of maximum depth $L$. At each iteration, the node representations from the previous iteration are updated, and the connectivity for the current depth is determined. Formally, let $x_i^l$ be the representation of the $i$-th node at depth $l$ for token $x$, with $l = 0, \ldots, L$. The DAG learning module produces the final token representation through the following procedure:

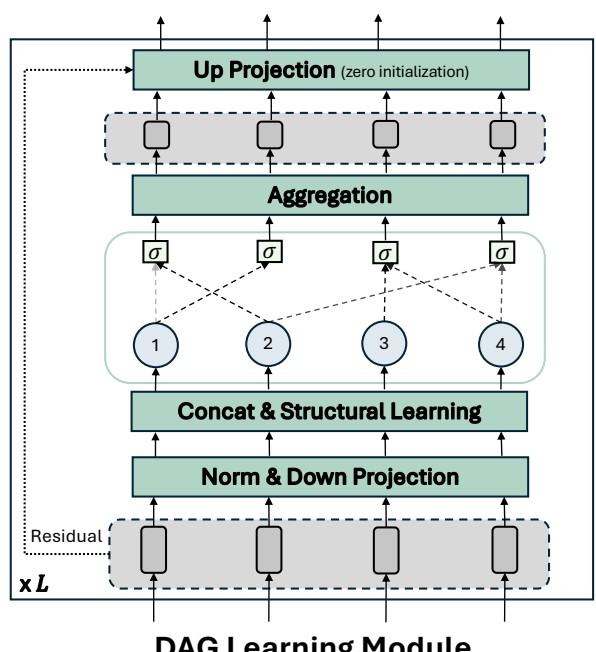

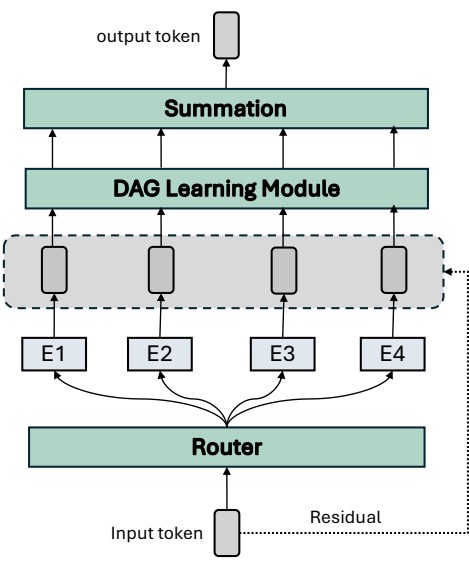

**DAG Learning Module**       **MoE Block**

*Figure 2.* Left: the DAG learning module automatically learns an optimal DAG structure over the selected experts and executes the DAG-style computation. Right: the complete MoE block in DAG-MoE.

$$x_i^0 = g_{\mathbf{k}[i]}(x)\, E_{\mathbf{k}[i]}(x) + \tfrac{1}{K}x, \quad i = 1, \ldots, K, \quad (6)$$

$$x_{i,\text{input}}^l = \text{LayerNorm}\left(x_i^{l-1}\right), \quad (7)$$

$$x_{i,\text{down}}^l = W_{\text{down}}^l x_{i,\text{input}}^l, \quad (8)$$

$$x_{(i,j)}^l = \text{Concat}(x_{i,\text{down}}^l, x_{j,\text{down}}^l), \quad (9)$$

$$e_{(i,j)}^l = \sigma\left(W_{\text{edge}}^l x_{(i,j)}^l\right), \quad \hat{x}_{(i,j)}^l = e_{(i,j)}^l * W_{\text{node}}^l x_{(i,j)}^l, \quad (10)$$

$$x_i^l = W_{\text{up}}^l \left(\sum_{j=1,\ldots,K} \hat{x}_{(i,j)}^l\right) + x_i^{l-1}, \quad (11)$$

where $W_{\text{down}} \in \mathbb{R}^{d_g \times d}$, $W_{\text{edge}} \in \mathbb{R}^{d_g \times 2d_g}$, $W_{\text{node}} \in \mathbb{R}^{d_g \times 2d_g}$, and $W_{\text{up}} \in \mathbb{R}^{d \times d_g}$ are learnable parameters, and $d_g$ is the hidden size of the DAG learning module, which can be set independently of both the model hidden size $d$ and the expert hidden size $d_r$. $\sigma$ is a nonlinear activation, implemented by the same activation function as in the expert FFN. The final output of the MoE block is obtained by summing the depth-$L$ node representations, which corresponds to taking $AGG$ as summation in the readout of Eq. 5:

$$y = \sum_{i=1}^{K} x_i^L. \quad (12)$$

We now detail the key design choices in the DAG learning module. First, Eq. 6 computes the output for each selected expert $i$, which serves as the initial node representation at depth 0. We additionally inject the original token representation through a residual connection; to keep the total residual contribution normalized to 1 after summing over all experts in Eq. 12, we scale each per-node residual by $1/K$. Empirically, both the residual connection and the scaling factor are crucial for training stability. To keep the DAG learning module lightweight, at each iteration we first normalize and then project the node representations into a lower-dimensional space via Eq. 7 and Eq. 8, and learn the structural relationships within this reduced space.

At iteration $l$, the structure is learned and executed as follows. First, we choose the number of nodes $n(l)$. In principle $n(l)$ can be any positive integer, leading to an enormous search space; in DAG-MoE we simply fix $n(l) = K$, matching the number of experts. Next, for each node $(l, i)$, $i = 1, \ldots, K$, we determine how it aggregates information from previous depths. Learning connections to all earlier nodes is expensive and yields overly dense graphs. To mitigate this, we restrict each node $(l, i)$ to aggregate only from nodes at depth $l - 1$, and inject information from earlier depths $0, \ldots, l - 2$ via residual connections. We use $x_{i,\text{down}}^l$ as the query for node $(l, i)$ and $\{x_{j,\text{down}}^l\}_{j=1}^K$ as the candidate keys, and learn the connection between $(l, i)$ and each $(l - 1, j)$ through Eq. 9–Eq. 10.

Concretely, we first form a candidate edge feature by concatenating $x_{i,\text{down}}^l$ with each $x_{j,\text{down}}^l$ for $j = 1, \ldots, K$. The

edge gate $e^l_{(i,j)}$ is then learned via the projection $W_{\text{edge}}$ followed by activation $\sigma$. This gate can encode various forms of relational information—for instance, logical operations in reasoning tasks, relations in knowledge graphs, or simply "no connection." The connection representation between $(l, i)$ and $(l - 1, j)$ is computed by Eq. 10 as an element-wise gating with $e^l_{(i,j)}$. Finally, the gated information is aggregated and projected back to the original dimension via Eq. 11, producing the output representation of node $(l, i)$. We use zero-weight initialization for the up-projection to stabilize early training. The complete workflow of the DAG learning module, together with the surrounding MoE block in DAG-MoE, is illustrated in Fig. 2.

**Computational cost analysis.** Since DAG-MoE introduces only the DAG learning module as an additional component, we focus our cost analysis on this module. Let the batch size be $B$ and the sequence length be $S$. The FLOPs of a single matrix multiplication are $2BSd_{\text{in}}d_{\text{out}}$. Based on this, the FLOPs of the DAG learning module for a single iteration are $\text{FLOPs}_{\text{dag}} = 4BSdd_g + 4K^2BS \cdot 2d_gd_g = 4BSdd_g + 8K^2BSd_g^2$. In comparison, the FLOPs for an additional shared expert in MoE with hidden dimension $d_g$ are $\text{FLOPs}_{\text{expert}} = 4BSdd_g + 2BSdd_g$. Subtracting and dividing out the common terms, the comparison reduces to $4K^2d_g$ versus $d$. In practice, since both $K \ll d$ and $d_g \ll d$, $4K^2d_g$ can be similar to or smaller than $d$. However, there will be additional overhead for the DAG learning module with multiple iterations due to its sequential nature.

# 4. Experiments

We empirically evaluate DAG-MoE with the goal of answering three questions: **Q1**: Does DAG-MoE outperform standard MoE across different base MoE configurations? **Q2**: How do the hyperparameters of the DAG learning module affect DAG-MoE's performance? **Q3**: After fine-tuning, how does DAG-MoE perform on downstream language tasks? To address these questions, we integrate DAG-MoE as the primary MoE block in an LLM and train the model from scratch on a causal language modeling objective. Our evaluation proceeds in two stages: a 12B-token study on smaller models to ablate the architectural and hyperparameter choices (Q1, Q2), followed by a 40B-token large-scale run with downstream fine-tuning that tests whether the architectural advantage transfers to end-task capabilities (Q3).

## 4.1. Experimental settings

We briefly summarize the pretraining and fine-tuning setup here, and defer the full details to Appendix B.

**Model details.** DAG-MoE follows the architecture of Llama3.1-8B (Dubey et al., 2024), retaining its tokenizer, attention module, and FFN design, while reducing the number of layers and the hidden size due to resource constraints. The MoE block uses a standard token-choice router following Switch Transformer (Fedus et al., 2022) with a load-balance loss; we additionally apply a router Z-loss to regularize the router logits (Muennighoff et al., 2025; Zoph et al., 2022). On top of the MoE block, we add the DAG learning module described in Section 3.4 and vary its hidden size $d_g$ and depth $L$. To systematically evaluate DAG-MoE across model scales, we design three variants: **DAG-MoE-s** (4 layers, hidden size 512), **DAG-MoE-m** (6 layers, hidden size 512), and **DAG-MoE-l** (8 layers, hidden size 1024). For the MoE block, DAG-MoE-s is studied under two granularity settings: (i) **coarse-grained**, with 32 experts of size $d_r = 256$ and top-4 routing; and (ii) **fine-grained**, with 64 experts of size $d_r = 128$ and top-8 routing. DAG-MoE-m and DAG-MoE-l use the coarse-grained setting only, with $d_r = 256$ and $d_r = 512$ respectively. As baselines, we use a standard MoE that shares the identical MoE configuration with DAG-MoE—same number of experts, same expert hidden size $d_r$, same top-$K$ router, and same training recipe—but without the DAG learning module. Since the DAG learning module introduces additional parameters, we further add a shared expert to the baseline whose hidden size is chosen so that the total parameter count exactly matches DAG-MoE, ensuring that any gain reflects structural aggregation rather than extra capacity.

**Data, training, and evaluation.** We use the Pile (Gao et al., 2020) as the pretraining corpus and design two setups. In the first, we train DAG-MoE-s and DAG-MoE-m on approximately 12B tokens randomly sampled from the Pile, and evaluate on a held-out 1.3B-token subset. In the second, we train DAG-MoE-l on about 40B tokens and evaluate on both in-domain (Pile) and out-of-domain corpora: FineWeb-Edu (Lozhkov et al., 2024), Wikipedia (Thrush et al., 2022), and C4 (Raffel et al., 2020). All models are trained with the causal language modeling objective at a maximum sequence length of 2048, with perplexity as the primary metric.

**Fine-tuning.** To evaluate downstream performance, we fine-tune both DAG-MoE-l and the corresponding baseline (each pretrained on 40B tokens) on a mixture of Alpaca (Taori et al., 2023), Open-Platypus (Lee et al., 2023), SlimOrca (Mukherjee et al., 2023), MathInstruct (Yue et al., 2023), Open-R1-Math[1], and MetaMathQA (Yu et al., 2023), for 3 epochs with a constant learning rate. We then evaluate on PIQA (Bisk et al., 2020), ARC-e (Clark et al., 2018), HellaSwag (Zellers et al., 2019), GPQA (Rein et al., 2024), Lambada (Paperno et al., 2016), MMLU (Hendrycks et al., 2021), and BBH (Suzgun et al., 2023). Detailed configurations and dataset descriptions are provided in Appendix B.2.

---

[1] https://huggingface.co/datasets/open-r1/OpenR1-Math-220k

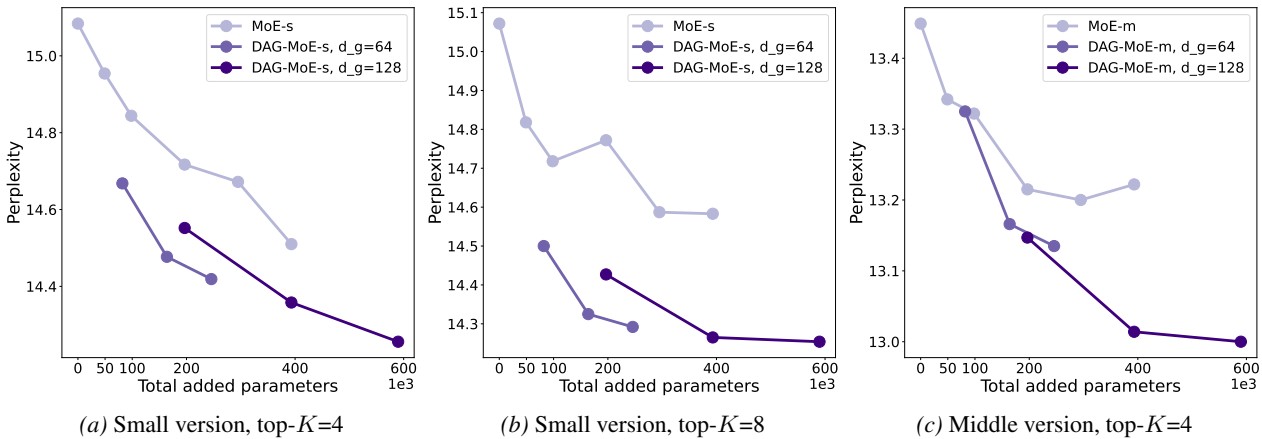

*(a)* Small version, top-$K$=4     *(b)* Small version, top-$K$=8     *(c)* Middle version, top-$K$=4

*Figure 3.* Perplexity of standard MoE and DAG-MoE on the Pile evaluation subset. The x-axis denotes the parameters added beyond the standard MoE block: for the baseline, the size of the added shared expert; for DAG-MoE, the product of the number of iterations $L$ and the per-iteration parameter count of the DAG learning module.

## 4.2. Pretraining evaluation results

We pretrain DAG-MoE-s, DAG-MoE-m, and their corresponding baselines on 12B tokens, varying the DAG learning module hyperparameters $d_g \in \{64, 128\}$ and $L \in \{1, 2, 3\}$. The main results are shown in Fig. 3. For the baseline MoE, we include variants both with and without a shared expert; the shared-expert size is encoded on the x-axis as "added parameters" (with $0$ denoting no shared expert). For DAG-MoE, the x-axis likewise indicates the parameters added by the DAG learning module: each curve corresponds to a fixed $d_g$, and each marker corresponds to a different $L \in \{1, 2, 3\}$, with the parameter count scaling linearly in $L$.

From Fig. 3, two observations stand out. First, **DAG-MoE consistently achieves lower perplexity than the standard MoE** across nearly all model sizes, granularities, and settings of $L$ and $d_g$. This empirically validates the expressiveness gain of structural aggregation: for a comparable parameter budget, the learned DAG affords greater flexibility in composing information from the selected experts. The improvement holds under both coarse-grained (top-$K$=4) and fine-grained (top-$K$=8) routing, suggesting that DAG-style aggregation is robust to MoE granularity. Second, even compared with a parameter-matched baseline that adds a shared expert, DAG-MoE attains a clear margin, indicating that the gain comes from the structural aggregation itself rather than from additional parameters alone. These observations directly answer **Q1**. We next dissect how the hyperparameters $L$ and $d_g$ contribute to this gain.

To answer **Q2**, Fig. 4 reports the perplexity improvement over the no-shared-expert baseline as a function of $L$. We make two observations. First, for both DAG-MoE-s and DAG-MoE-m and a fixed $d_g$, performance improves as $L$ increases, with the largest gains coming from $L = 0 \rightarrow$

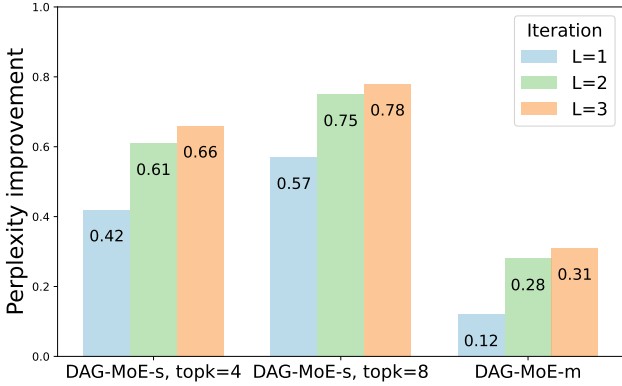

*Figure 4.* Perplexity reduction of DAG-MoE (with $d_g = 64$) over the no-shared-expert MoE baseline as a function of the number of DAG iterations $L$. Higher is better.

1 and $L = 1 \rightarrow 2$; for example, on DAG-MoE-s with both top-$K$=4 and top-$K$=8, a single iteration with $d_g$=64 already yields about a $0.5$ reduction in perplexity. The improvement from $L = 2$ to $L = 3$ is marginal, suggesting that one or two iterations already suffice to capture the relevant compositional structure for most tokens. Second, increasing $d_g$ also helps but is less effective per parameter than increasing $L$: in Fig. 3a, DAG-MoE with $d_g$=64, $L$=2 outperforms $d_g$=128, $L$=1 despite adding fewer parameters, and the same trend holds for the fine-grained (Fig. 3b) and medium-scale (Fig. 3c) settings.

*Table 1.* Pretraining perplexity ($\downarrow$) of DAG-MoE-l vs. MoE-l on in-domain (Pile) and out-of-domain (Wiki, FineWeb-Edu, C4) corpora. Both models have 699M parameters. Best per column in bold.

| Perplexity $\downarrow$ | Pile | Wiki | FineWeb | C4 |
|---|---|---|---|---|
| MoE-l | 10.51 | 21.08 | 25.38 | 35.21 |
| DAG-MoE-l | **10.27** | **20.54** | **24.69** | **34.21** |

To further validate the gain at scale, we pretrain DAG-MoE-

l on 40B tokens. Here DAG-MoE-l uses a DAG learning module with $d_g = 256$ and $L = 2$, and the baseline MoE-l adds a shared expert with $d_r = 512$, so that both models have 699M parameters. We evaluate perplexity on both the in-domain corpus (Pile) and three out-of-domain corpora (Wikipedia, FineWeb-Edu, C4). As shown in Table 1, DAG-MoE-l consistently outperforms MoE-l under the same training and parameter budget, with the gap noticeably larger on out-of-domain corpora ($-0.69$ on FineWeb-Edu and $-1.00$ on C4, vs. $-0.24$ on Pile). A natural reading is that out-of-domain text places higher demand on the model's ability to compose information flexibly across experts, so the expressiveness advantage formalized in Theorem 3.2 surfaces more visibly there. We provide the training curves and additional discussion in Appendix C.1.

### 4.3. Comparison with alternative designs and efficiency

We further ask whether the gain is specific to the *DAG-style* structural aggregation and what wall-clock cost it incurs. We address both with controlled ablations on DAG-MoE-s under the coarse-grained setting (top-$K$=4, 12B tokens).

*Table 2.* Comparison with CoE on DAG-MoE under matched added-parameter budget. $\Delta$ PPL is the perplexity reduction over the standard MoE.

| Model | Add. Params | $\Delta$ PPL $\uparrow$ |
|---|---|---|
| Standard MoE | 0 | 0.000 |
| + shared expert | 393K | 0.433 |
| CoE | 393K | 0.480 |
| DAG-MoE-s | 393K | **0.587** |

**Comparison with chain-of-experts.** Among existing architectures, Chain-of-Experts (CoE) (Wang et al., 2025) is the closest to DAG-MoE in spirit, as it also processes selected experts iteratively within a single MoE layer. We reimplement CoE under our codebase and match its added-parameter budget to DAG-MoE (shared expert $d_g$=256; CoE $L$=2, $d_g$=256; DAG-MoE-s $L$=2, $d_g$=128; all add 393K parameters per layer). As shown in Table 2, DAG-MoE attains a clearly larger perplexity reduction than CoE (0.587 vs. 0.480), indicating that the gain comes from the *structural form* of aggregation rather than iterative refinement alone.

*Table 3.* Replacing the DAG learning module with naive MLP mixing on DAG-MoE-s.

| Model | Add. Params | Eval Loss $\downarrow$ |
|---|---|---|
| Standard MoE | 0 | 2.7168 |
| + shared expert ($d_g$=32) | 49K | 2.7072 |
| + shared expert ($d_g$=64) | 98K | 2.7012 |
| MLP mixing ($d_g$=32) | 49K | 2.8406 |
| MLP mixing ($d_g$=64) | 98K | 2.8006 |
| DAG-MoE-s ($d_g$=32, $L$=1) | 37K | 2.6948 |
| DAG-MoE-s ($d_g$=32, $L$=2) | 74K | **2.6899** |

**Replacing the DAG with naive MLP mixing.** A second natural alternative is to drop the DAG entirely and concatenate the selected expert outputs into a single MLP with hidden size matched to the added-parameter budget. As shown in Table 3, this naive MLP mixing in fact *degrades* performance below the standard MoE at both budgets, despite using strictly more parameters than DAG-MoE. The contrast suggests that the DAG provides an inductive bias—structured, iterative composition of experts—that an unconstrained mixer cannot recover from data on a fixed MoE budget.

*Table 4.* Training throughput analysis of DAG-MoE-s vs. the standard MoE. Tok/s ratios relative to the standard MoE in parentheses.

| Model | Params | FLOPs (E) | Tok/s (M) |
|---|---|---|---|
| Std. MoE | 184.75M | 8.931 | 0.7744 (1.000) |
| DAG-MoE-s ($L$=1) | 184.69M | 8.926 | 0.7627 (0.985) |
| DAG-MoE-s ($L$=2) | 185.02M | 8.951 | 0.7404 (0.956) |

**Throughput overhead.** Table 4 reports the wall-clock training throughput of DAG-MoE-s vs. the standard MoE under identical hardware and data-loading conditions. DAG-MoE adds only 1.51% overhead at $L$=1 and 4.49% at $L$=2, with essentially identical total parameters and FLOPs—a modest cost for the perplexity gains shown above, and one that could be further reduced via standard kernel-level optimizations (e.g., `torch.compile`). To complement these quantitative results, we visualize the learned DAG structure—both the mean edge weights at each (layer, iteration) and the per-token spread under t-SNE—in Appendix C.2, which shows that DAG-MoE learns diverse, layer-specific, and token-adaptive aggregation patterns rather than collapsing to a single fixed graph.

### 4.4. Fine-tuning evaluation results

The pretraining results above establish that DAG-MoE reduces perplexity over standard MoE under matched parameter and compute budgets. We now examine **Q3**: whether this architectural advantage transfers to actual end-task capability after instruction fine-tuning. We fine-tune the pretrained DAG-MoE-l and MoE-l on the instruction datasets above (under identical configurations) and evaluate on the seven downstream benchmarks. The results are shown in Table 5. DAG-MoE-l outperforms or matches the baseline on six of seven benchmarks, with an average accuracy of 26.13 versus 24.06. The gains concentrate on tasks that demand multi-step inference: DAG-MoE-l improves over MoE-l by $+6.06$ on GPQA, $+3.46$ on Lambada, $+3.15$ on PIQA, and $+0.90$ on BBH, while remaining within noise on more pattern-matching benchmarks such as HellaSwag and MMLU. This pattern is consistent with the structural-aggregation motivation underlying DAG-MoE: by composing expert outputs through a learned DAG rather than a permutation-invariant sum, DAG-MoE is better positioned to support the hierarchical, ordered aggregation that compositional reasoning

*Table 5.* Downstream accuracy (↑) of DAG-MoE-l vs. MoE-l after fine-tuning on the same instruction-tuning mixture. HellaSwag is evaluated 10-shot, MMLU 5-shot, others 0-shot. Best per column in bold.

| Accuracy ↑ | PIQA | ARC-e | HellaSwag | GPQA | Lambada | MMLU | BBH | Average |
|---|---|---|---|---|---|---|---|---|
| MoE-l | 47.52 | 24.34 | **25.90** | 21.72 | 8.11 | **24.17** | 16.65 | 24.06 |
| DAG-MoE-l | **50.67** | **25.57** | 25.73 | **27.78** | **11.57** | 24.03 | **17.55** | **26.13** |

benefits from—without claiming, of course, that the learned DAG literally implements any specific algorithmic procedure (cf. the remarks following Theorem 3.3). Combined with the perplexity gains in pretraining, these results suggest that the expressiveness benefit of structural aggregation is not merely a property of the language-modeling loss but is also realized in downstream behavior, particularly on tasks that reward compositional inference.

## 5. Related Works

**Mixture of Experts (MoEs).** MoEs (Shazeer et al., 2017; Fedus et al., 2022) have become a dominant paradigm for large-scale models such as LLMs, as they decouple parameter count from per-token compute and thus allow capacity to scale without a proportional increase in inference cost. Many state-of-the-art LLMs adopt MoE architectures, including Mixtral (Jiang et al., 2024), DeepSeek-V3 (Liu et al., 2024), and Qwen-3 (Yang et al., 2025).

**Architecture improvements of MoEs.** Several works modify the core MoE design to enhance its capacity. MH-MoE (Huang et al., 2024) splits each input into multiple "heads" that route independently to experts, analogous to multi-head attention. CoE (Wang et al., 2025) introduces an iterative routing strategy that selects top-$K$ experts over multiple rounds and refines the output step by step; this requires an independent router at each stage, so the routing cost grows linearly with the number of iterations, whereas DAG-MoE routes only once and adds structural aggregation as a lightweight extension. DiEP (Bai et al., 2026) also models an MoE layer with a DAG, but uses it as a differentiable search space for *expert pruning*, targeting model compression rather than expert aggregation. Most related to our work, S′MoRE (Zeng et al., 2025) introduces structural flexibility into MoE through a hierarchical router that selects experts across multiple stages, with experts at adjacent stages connected by a non-linear transformation, forming a tree-shaped computation. S′MoRE, however, is designed as a parameter-efficient fine-tuning adapter rather than a standalone MoE backbone, and its structural form is fixed to a predefined tree. In contrast, DAG-MoE treats expert composition as a general DAG and learns a token-adaptive structure end-to-end, generalizing prior structural variants beyond a fixed hierarchy.

**Scaling and routing of MoEs.** A growing body of work studies and improves MoE scaling. OLMoE (Muennighoff et al., 2025) and Skywork-MoE (Wei et al., 2024) conduct extensive empirical studies on the components and training strategies of MoE. On the routing side, several works propose alternative routing algorithms, including expert-choice routing (Zhou et al., 2022) and RNN-based routing (Qiu et al., 2024); others analyze the representation gap between router and token embeddings (Chi et al., 2022), or improve training stability by refining the load-balance loss in token-choice routing (Wang et al., 2024; Qiu et al., 2025). On the scaling-law side, Abnar et al. (2025) characterizes the optimal sparsity level for MoE, Ludziejewski et al. (2025) jointly studies data, model, and training strategy, and Tian et al. (2025) develops scaling laws for efficient MoE training. Granularity-oriented scaling (He, 2024; Ludziejewski et al., 2024), the closest counterpart to DAG-MoE's expressiveness axis, is complementary to the structural aggregation introduced in this work.

## 6. Conclusions and Limitations

In this paper, we replace the simple weighted summation in MoE with structural aggregation. By formulating expert aggregation as a DAG, we enlarge the space of expert combinations and enhance flexibility without modifying the router or expert configurations. To this end, we propose DAG-MoE, which dynamically learns an optimal DAG structure via a lightweight DAG-learning module. Across both language modeling and downstream tasks, DAG-MoE consistently outperforms standard MoE, confirming its effectiveness. The current implementation, however, restricts the class of DAGs (e.g., a fixed number of nodes per depth and depth-adjacent connections), so only Theorem 3.2 fully transfers, while Proposition 3.1 applies to the realizable subclass and Theorem 3.3 should be read as motivation rather than a guarantee. Identifying the optimal DAG structure for a given token—and how to learn it effectively within the module—also remains underexplored. Finally, our evaluation is limited to small-scale training, and it is unclear how DAG-MoE would perform at billion-parameter, trillion-token scale. We leave these directions to future work.

## Impact Statement

This paper presents work whose goal is to advance the field of Machine Learning. There are many potential societal consequences of our work, none of which we feel must be specifically highlighted here.

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

# A. Detailed proofs of all theorems

## A.1. Detailed proofs for expressiveness of DAG-MoE

For completeness, we first restate the general formulation of DAG-style MoE. Let $x_i^l$ denote the output of node $v = (l, i)$. For a given DAG $G \in \mathcal{G}(K)$, the corresponding computation in DAG-style MoE can be formulated as:

$$x_i^0 = g_{\mathbf{k}[i]}(x) E_{\mathbf{k}[i]}(x), \quad i = 1, \ldots, K, \tag{13}$$

$$x_i^l = AGG(\{x_j^k \mid (k, j) \in A_i^l\}), \quad i = 1, \ldots, n(l), \ l = 1, \ldots, L-1, \tag{14}$$

$$y = AGG(\{x_j^k \mid (k, j) \in A_1^L\}). \tag{15}$$

We leverage results from D-VAE (Zhang et al., 2019), which encodes computations over a DAG via an injective aggregation–update scheme executed in topological order. Concretely, for a DAG $G = (\mathcal{V}, \mathcal{A})$ with an initial node $(0, 0)$, D-VAE encodes the DAG via:

$$x_0^0 = \hat{x}_0^0, \tag{16}$$

$$x_i^l = AGG(\{U(\hat{x}_i^l, x_k^j) \mid (k, j) \in A_i^l\}), \quad i = 1, \ldots, n(l), \ l = 1, \ldots, L-1, \tag{17}$$

$$y = AGG(\{x_j^k \mid (k, j) \in A_1^L\}). \tag{18}$$

where $AGG$ (aggregation) and $U$ (node update) are injective, and $\hat{x}_i^l$ is the initial feature of node $(l, i)$. Zhang et al. (2019) provide the following conclusion, which we restate here:

**Proposition A.1.** *Let $G$ be a DAG with a single initial node $(0, 0)$. Eq. 16-Eq. 18 can map $G$ to $y$ injectively if $AGG$ and $U$ are injective.*

*Proof.* See Theorem 2 in D-VAE (Zhang et al., 2019). □

With Proposition A.1 in place, we are ready to prove Proposition 3.1, which we restate here:

**Proposition A.2.** *Given a top-$K$ expert list $\mathbf{k}$, any DAG-style MoE satisfying Eq. 3-Eq. 5 can injectively encode any $G \in \mathcal{G}(K)$ if $AGG$ is injective.*

*Proof.* To prove the proposition, we reduce Eq. 13–Eq. 15 to a special case of the D-VAE encoder in Eq. 16–Eq. 18. Before doing so, we note several differences between the definitions in DAG-style MoE and D-VAE.

- In D-VAE, any DAG $G$ is assumed to have a single start node $(0, 0)$. Here, however, we have $K$ start nodes, each corresponding to the output of one expert. To align with D-VAE, we observe that all representations are generated from the input token $x$, so $x$ itself can serve as node $(0, 0)$ to match Eq. 16. Then the previous $K$ start nodes become nodes at iteration 1, and so on.

- In D-VAE, each node in $G$ is assumed to have an initial feature $\hat{x}_i^l$ that is integrated via the injective $U$ function, whereas DAG-style MoE has no such per-node initial feature. To align DAG-style MoE with D-VAE, we assume that every node in $G$ has the same $\mathbf{0}$-initialized input feature, except for node $(0, 0)$. This modification does not affect the expressiveness of DAG-style MoE since no initial features are used in the original formulation.

Given these changes, we rewrite the formulation of DAG-style MoE as follows:

$$x_0^0 = x, \tag{19}$$

$$x_i^1 = g_{\mathbf{k}[i]}(x_0^0) E_{\mathbf{k}[i]}(U(\mathbf{0}, x_0^0)), \quad i = 1, \ldots, K, \tag{20}$$

$$x_i^l = AGG(\{U(\mathbf{0}, x_j^k) \mid (k, j) \in A_i^l\}), \quad i = 1, \ldots, n(l), \ l = 2, \ldots, L, \tag{21}$$

$$y = AGG(\{x_j^k \mid (k,j) \in A_1^{L+1}\}). \tag{22}$$

The above equations almost coincide with the D-VAE encoder, with a few differences. First, we add $U$ to DAG-style MoE to match the D-VAE equation; however, since the initial feature here is $\mathbf{0}$, this does not affect the expressiveness of the model, and $U$ can be absorbed into $AGG$ to form a new aggregation function. Second, at the first iteration, every node has exactly one incoming predecessor under our DAG definition, so using $E(\cdot)$ as the $AGG$ function is equivalent to the original formulation, since each node aggregates from a singleton parent. Meanwhile, $g_i$ and $E_i$ can easily be implemented as injective functions. Consequently, Eq. 20 can be further rewritten as:

$$x_i^1 = AGG(\{U(\mathbf{0}, x_0^0)\}), \quad i = 1, \dots, K. \tag{23}$$

Now the DAG-style MoE process matches D-VAE exactly, and the proposition follows directly from Proposition A.1. We omit further details. □

Next, we prove Theorem 3.2, which we restate here:

**Theorem A.3.** *Given a top-$K$ expert list $\mathbf{k}$, there exists a DAG-style MoE satisfying Eq. 13–Eq. 15 that is strictly more powerful than the standard MoE in Eq. 1, provided that $AGG$ is injective.*

*Proof.* First, it is straightforward that DAG-style MoE is at least as powerful as standard MoE. Set $G$ to have $K$ isolated nodes (no interactions among the top-$K$ experts; see the left panel of Fig. 1) and take $L = 0$. Then Eq. 14 vanishes and Eq. 15 reduces to a readout over $\{x_i^0 = g_i(x)E_i(x)\}_{i=1}^K$. With AGG chosen as summation, this exactly recovers the standard MoE weighted sum.

Next, we show that DAG-style MoE can map different DAG structures to different outputs, whereas standard MoE always produces the same representation for a fixed set of selected experts. Consider the tree in the middle of Fig. 1 and define two DAGs that differ only in how two leaves are paired at the first depth:

$$G_1 = (\mathcal{V}_1, \mathcal{A}_1), \quad \mathcal{A}_1 = \{A_1^1 = \{(0,1),(0,2)\}, \ A_2^1 = \{(0,3),(0,4)\}, \ A_1^2 = \{(1,1),(1,2)\}\},$$

$$G_2 = (\mathcal{V}_2, \mathcal{A}_2), \quad \mathcal{A}_2 = \{A_1^1 = \{(0,1),(0,3)\}, \ A_2^1 = \{(0,2),(0,4)\}, \ A_1^2 = \{(1,1),(1,2)\}\},$$

Let $x_{g,i}^0$ denote the initial representation of node $(0,i)$ in graph $G_g$. Assume the four leaf representations are the same in both graphs, i.e., $x_{1,i}^0 = x_{2,i}^0$ for $i = 1, \dots, 4$, and that $x_{\cdot,2}^0 \neq x_{\cdot,3}^0$. Then the predecessor multisets at depth 1 differ between $G_1$ and $G_2$ (one pairs $(0,1)$ with $(0,2)$, the other pairs $(0,1)$ with $(0,3)$); hence the depth-1 node states differ because AGG is injective on multisets. By composition of injective maps, all downstream states—and therefore the final outputs—also differ. Formally, by Proposition A.2, DAG-style MoE maps $G_1$ and $G_2$ to distinct outputs. In contrast, standard MoE aggregates the same leaf set $\{x_{\cdot,i}^0\}_{i=1}^4$ by a permutation-invariant weighted sum, yielding identical outputs for $G_1$ and $G_2$. Thus, DAG-style MoE strictly separates these two structures while standard MoE does not, which concludes the proof. □

### A.2. Detailed discussion on dynamic programming

#### A.2.1. DEFINITION OF DYNAMIC PROGRAMMING

In this section, we formally define the dynamic programming problem following previous work (Feng et al., 2023). A general DP algorithm can be characterized via three key ingredients: a state space $\mathcal{I}$, a transition function $T$, and an aggregation function $AGG$. Given a DP problem with $N$ input sequences $\boldsymbol{s}^{(1)}, \cdots, \boldsymbol{s}^{(N)}$, we denote the problem size as the vector $\boldsymbol{n} = (|\boldsymbol{s}^{(1)}|, \cdots, |\boldsymbol{s}^{(N)}|)$. Given the fixed problem size $\boldsymbol{n}$, there is an associated state space $\mathcal{I}_{\boldsymbol{n}} \subset \mathcal{I}$ representing the finite set of decomposed subproblems, where each state $i \in \mathcal{I}_{\boldsymbol{n}}$ is an index signifying a specific subproblem. The size of the state space $\mathcal{I}_{\boldsymbol{n}}$ grows with the problem size $\boldsymbol{n}$. We denote by $dp(i)$ the answer along with other information about the DP process of subproblem $i$. Furthermore, there is a partial order between different states: we say state $j$ precedes state $i$ (denoted as $j \prec i$) if subproblem $j$ must be solved before subproblem $i$. This partial order naturally induces a DAG over the state space, thereby establishing a reasoning chain that can be approximated by DAG-style MoE.

In the paper, we focus on a restricted setting where each state $i$ only depends on (i) a finite number of tokens in the input sequence $\boldsymbol{s}$ and (ii) a finite number of previous states. Under this assumption, the transition function $T$ can be generally written as:

$$
\begin{aligned}
dp(i) &= f(\boldsymbol{n}, i, s_{\boldsymbol{g}(\boldsymbol{n},i)}, dp(\boldsymbol{h}(\boldsymbol{n},i))) \\
&= f(\boldsymbol{n}, i, s_{g_1(\boldsymbol{n},i)}, \cdots, s_{g_J(\boldsymbol{n},i)}, dp(h_1(\boldsymbol{n},i)), \cdots, dp(h_P(\boldsymbol{n},i))),
\end{aligned}
\tag{24}
$$

where the functions $f, \boldsymbol{g}, \boldsymbol{h}$ fully determine the transition function $T$ and have the following forms: $f : \mathbb{N}^N \times \mathcal{I} \times \mathcal{X}^J \times \mathcal{Y}^P \to \mathcal{Y}$, $\boldsymbol{g} : \mathbb{N}^N \times \mathcal{I} \to (\mathbb{N} \cup \{\emptyset\})^J$, and $\boldsymbol{h} : \mathbb{N}^N \times \mathcal{I} \to (\mathcal{I} \cup \{\emptyset\})^P$. Here, the state space $\mathcal{I}$, input space $\mathcal{X}$, and DP output space $\mathcal{Y}$ can be arbitrary domains, and $J, P$ are constant integers. If state $i$ depends on fewer than $J$ input tokens or fewer than $P$ previous states, we use the special symbol $\emptyset$ as a placeholder, so that all terms $s_\emptyset$ and $dp(\emptyset)$ are unused in $f$. Concretely, $\boldsymbol{g}(\boldsymbol{n}, i)$ indicates the input indices used to compute the transition $dp(i)$, while $\boldsymbol{h}(\boldsymbol{n}, i)$ indicates the previous DP states required. After solving all subproblems, the aggregation function $AGG$ is used to combine all results and obtain the final answer. We consider a general class of aggregation functions with the following form:

$$
AGG(\{(i, dp(i)) : i \in \mathcal{I}_{\boldsymbol{n}}\}) = u(\square_{i \in \mathcal{A}_{\boldsymbol{n}}} dp(i)),
\tag{25}
$$

where $\mathcal{A}_{\boldsymbol{n}} \subset \mathcal{I}_{\boldsymbol{n}}$ is the set of states to aggregate, $\square$ is an aggregation operator such as $\min$, $\max$, or $\sum$, and $u : \mathcal{Y} \to \mathcal{Z}$ is any function where $\mathcal{Z}$ denotes the space of possible answers. Next, we describe how to construct the DAG induced by a DP problem.

**Definition A.4.** Given a DP problem with input sequences $\boldsymbol{s}^{(1)}, \cdots, \boldsymbol{s}^{(N)}$, problem size $\boldsymbol{n} = (|\boldsymbol{s}^{(1)}|, \cdots, |\boldsymbol{s}^{(N)}|)$, total input length $|\boldsymbol{n}| = \sum_t |\boldsymbol{s}^{(t)}|$, and the transition function specified in Eq. 24, we define the DAG $G_{dp}$ induced by the DP solving process as follows:

- There are $|\boldsymbol{n}|$ *input nodes* at depth 0, indexed by input position $j = 1, \ldots, |\boldsymbol{n}|$, where node $(0, j)$ stores the input token $s_j$.

- For each depth $l > 0$, each *subproblem node* $(l, i)$ corresponds to a subproblem $i \in \mathcal{I}_{\boldsymbol{n}}$ and represents the solution $dp(i)$, such that for every $j \in \boldsymbol{h}(\boldsymbol{n}, i)$ the depth of node $(\cdot, j)$ is less than $l$.

- The last depth $L(dp)$ contains a single *output node* $(L(dp), 1)$ representing the final answer $y$ of the DP problem.

- The adjacency list is $\mathcal{A} = \{A_i^l\}$, where for each interior subproblem node,

$$
A_i^l = \{(d(j), j) \mid j \in \boldsymbol{h}(\boldsymbol{n}, i)\} \cup \{(0, j) \mid j \in \boldsymbol{g}(\boldsymbol{n}, i)\}, \quad l = 1, \ldots, L(dp) - 1,
$$

  with $d(j)$ denoting the depth of subproblem node $(\cdot, j)$.

- For the output node, $A_1^{L(dp)} = \{(d(i), i) \mid i \in \mathcal{A}_{\boldsymbol{n}}\}$, where $\mathcal{A}_{\boldsymbol{n}} \subseteq \mathcal{I}_{\boldsymbol{n}}$ is the set of subproblems aggregated by the final readout.

Two elements warrant further discussion. First, $d(i)$ is the depth of subproblem $i$ in the DAG, defined as the smallest iteration at which $dp(i)$ can be obtained—i.e., the smallest iteration at which all $dp(j)$ for $j \in \boldsymbol{h}(\boldsymbol{n}, i)$ are ready. Second, $L(dp)$ is the depth of the DAG $G_{dp}$, i.e., the smallest iteration at which the final DP answer is produced.

### A.2.2. LONGEST INCREASING SUBSEQUENCE PROBLEM AND AN EXAMPLE ON HOW DAG-MOE CAN SIMULATE ITS SOLVING PROCESS

In this section, we describe one representative DP problem: the longest increasing subsequence (LIS) problem. The LIS problem aims to compute the length of the longest increasing subsequence of an input sequence $\mathbf{s} \in \mathbb{N}^n$. We say $\tilde{\mathbf{s}}$ is a subsequence of $\mathbf{s}$ if there exist indices $1 \leq i_1 \leq \cdots \leq i_{|\tilde{\mathbf{s}}|} \leq n$ such that $\tilde{s}_k = s_{i_k}$ for all $k = 1, \ldots, |\tilde{\mathbf{s}}|$. A sequence $\tilde{\mathbf{s}}$ is called increasing if $\tilde{s}_1 \leq \cdots \leq \tilde{s}_{|\tilde{\mathbf{s}}|}$. The LIS problem aims to find an increasing subsequence of $\mathbf{s}$ of maximal length. Let $h(n, i) = \{j \mid s_j < s_i\}$ be the predecessor index set, including all element indices $j$ such that $s_j < s_i$. The transition function of the LIS problem is:

$$
dp(i) = 1 + \max(\{dp(j) \mid j \in h(n, i)\}), \quad \max(\emptyset) = 0.
\tag{26}
$$

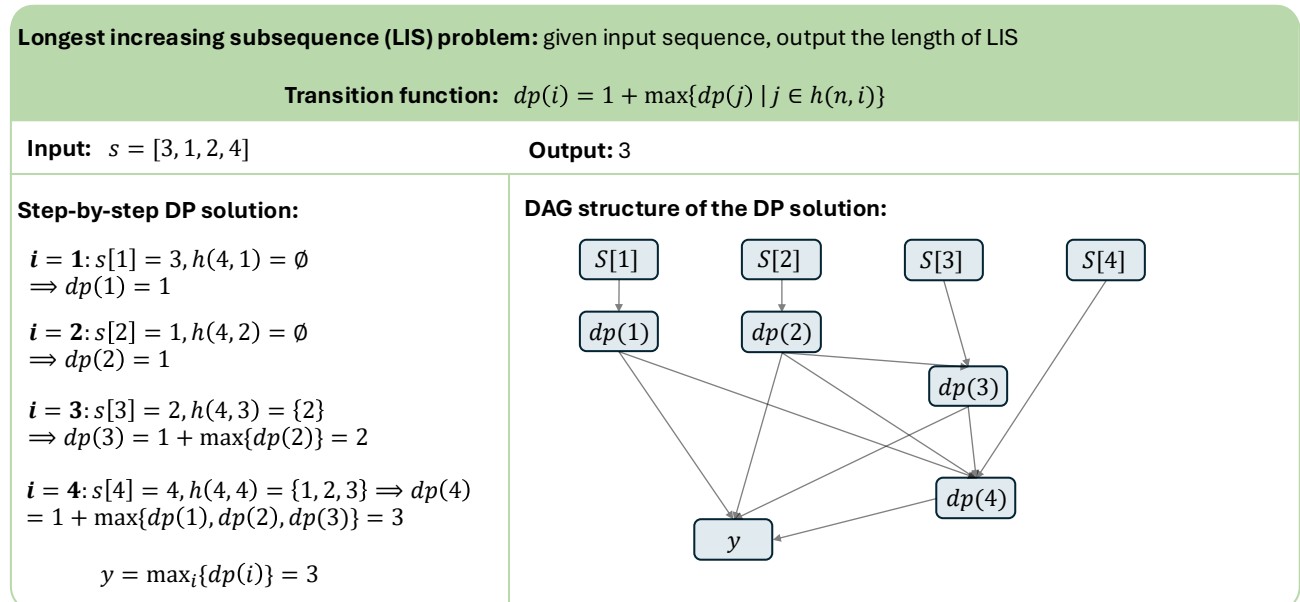

**Longest increasing subsequence (LIS) problem:** given input sequence, output the length of LIS

**Transition function:** $dp(i) = 1 + \max\{dp(j) \mid j \in h(n, i)\}$

**Input:** $s = [3, 1, 2, 4]$ — **Output:** 3

**Step-by-step DP solution:**

$i = 1$: $s[1] = 3, h(4, 1) = \emptyset$
$\Rightarrow dp(1) = 1$

$i = 2$: $s[2] = 1, h(4, 2) = \emptyset$
$\Rightarrow dp(2) = 1$

$i = 3$: $s[3] = 2, h(4, 3) = \{2\}$
$\Rightarrow dp(3) = 1 + \max\{dp(2)\} = 2$

$i = 4$: $s[4] = 4, h(4, 4) = \{1, 2, 3\} \Rightarrow dp(4)$
$= 1 + \max\{dp(1), dp(2), dp(3)\} = 3$

$$y = \max_i\{dp(i)\} = 3$$

**DAG structure of the DP solution:**

*Figure 5.* An example of the LIS problem and the corresponding DAG structure.

The final solution can be obtained by:

$$y = \max_i(\{dp(i) \mid i = 1, \ldots, n\}) \tag{27}$$

Here we show a small example with the sequence $[3, 1, 2, 4]$ at the top of Fig. 5. By iteratively applying the transition function, we obtain the final answer 3, as illustrated on the left of Fig. 5. At the same time, the process can be converted into a DAG that specifies the computation, as shown on the right of Fig. 5.

Now, suppose we have a generic DAG-style MoE (in the sense of Equations 13–15) that selects top-4 experts and whose adjacency $\mathcal{A}$ matches $G(dp)$ for the LIS instance in Fig. 5. We feed the sequence into the model and query the answer at the [ANS] position. In the first attention block, the attention module gathers the entire input sequence into the [ANS] token (Block 1 of Theorem A.5). At the MoE block, each expert is implemented as a linear projection that selects one element of the input sequence, forming the depth-0 leaves of $G(dp)$. Expert outputs are then aggregated according to $G(dp)$ to produce the final answer. By Assumption 4.3 of Feng et al. (2023), $AGG_{\text{trans}}$ can approximate the LIS transition $dp(i) = 1 + \max(\{dp(j) \mid j \in h(n, i)\})$ at each interior node, and $AGG_{\text{out}}$ at the readout approximates $y = \max_i dp(i)$. Thus, in principle, a single DAG-style MoE layer can simulate the entire LIS solving process in Fig. 5.

### A.3. Proof of Theorem 3.3

With the definitions above in place, we now prove Theorem 3.3. The theorem is built upon Theorem 4.7 of Feng et al. (2023) and inherits Assumptions 4.2–4.5 of Feng et al. (2023); we refer the reader to the original paper for the formal statements of these assumptions. We restate Theorem 3.3 below:

**Theorem A.5.** *For any integer $n \in \mathbb{N}$, and any DP problem satisfying Assumptions 4.2–4.5 of Feng et al. (2023) with corresponding DAG $G(dp)$ and total input length $|\mathbf{n}|$ at most $O(K \log n)$, there exists a log-precision Transformer consisting of (i) one multi-head attention layer with $H = O(K \log n)$ heads and (ii) one generic DAG-style MoE block (Equations 13–15) with top-$K$ experts, $L \geq L(dp)$ iterations sharing the same parameters, and adjacency $\mathcal{A}$ specified by $G(dp)$, with hidden dimension $d = O(K \log n)$ and per-iteration parameter count $O(\text{poly}(K))$, such that all parameter values are bounded by $O(\text{poly}(n))$ and the output token is the correct DP answer.*

*Proof.* We first highlight the differences between Theorem A.5 and Theorem 4.7 of Feng et al. (2023). Theorem 4.7 considers DP problems of input length up to $n$ and shows that an autoregressive Transformer can solve them step by step via a Chain-of-Thought mechanism: by iteratively decoding, it produces an $O(\text{poly}(n))$-length sequence storing all input tokens and intermediate DP states, and uses attention to retrieve the information needed at each step. In Theorem A.5, instead, a single DAG-style MoE block simulates the DP solving process within one forward pass via the DAG itself,

without intermediate decoding. The trade-off is that we only have $K$ experts, all derived from a single input token. Under log-precision (see (Feng et al., 2023) for the formal definition), each scalar can store $O(\log n)$ bits, so a single token of dimension $d$ can store $O(d \log n)$ bits in total. Setting $d = O(K \log n)$ thus permits storing $O(K \log n)$ input tokens in one representation, which constrains the input length to $|\boldsymbol{n}| = O(K \log n)$. Below we describe how to construct the two blocks.

**Input format.** We represent the DP instance as a sequence of tokens:

$$\boldsymbol{s}^{(1)} \mid \cdots \mid \boldsymbol{s}^{(N)} \mid [\text{ANS}],$$

where $\boldsymbol{s}^{(t)}$ is the $t$-th input sequence and $[\text{ANS}]$ is a designated query position at which the model emits the DP answer.

**Block 1 (gather inputs into a single token).** The first block uses one multi-head attention layer with $H = |\boldsymbol{n}| = O(K \log n)$ heads to gather all input tokens into the representation at the $[\text{ANS}]$ position. We allocate one attention head per input position $j \in [|\boldsymbol{n}|]$: head $j$ uses position-based query and key vectors so that, at the $[\text{ANS}]$ position, the query inner product is maximized exclusively at key position $j$; the value at position $j$ is the input token embedding $s_j$, which the head writes into a dedicated $O(\log n)$-dimensional slice of the output. Each head thus realizes a positional COPY operation in the sense of Lemma C.7 of Feng et al. (2023), and the simultaneous use of multiple heads to copy several token-indexed sources into different output slices follows the same pattern as Block 3 in the proof of Theorem 4.7 of Feng et al. (2023), which uses $K + J$ heads to copy $K + J$ token-indexed embeddings in parallel. The only departure from the setting of Feng et al. (2023) is that here the number of heads $H$ scales with input length rather than being constant, which is the unavoidable cost of collapsing the entire DP execution into a single forward pass without intermediate decoding. After this block, the representation at the $[\text{ANS}]$ position contains all input tokens of $\boldsymbol{s}^{(1)}, \ldots, \boldsymbol{s}^{(N)}$ concatenated across slices, occupying $O(K \log n)$ dimensions in total.

**Block 2 (DAG-style MoE simulates $G(dp)$).** Given the gathered token representation, we use one generic DAG-style MoE block (Equations 13–15) to execute the DP. We hard-code the adjacency $\mathcal{A}$ to coincide with $G(dp)$ (Definition A.4); this is admissible because, in the generic formulation, $\mathcal{A}$ is a structural specification rather than learnable. By Assumption 4.4 of Feng et al. (2023), there is a feasible topological ordering of $\mathcal{I}_{\boldsymbol{n}}$, so we may identify each DP subproblem $i$ with a node $(d(i), i)$ in the DAG-MoE block at the depth $d(i)$ given by Definition A.4.

We construct the experts and aggregation as follows. Each of the $K$ experts is implemented as a linear projection that selects $\lceil |\boldsymbol{n}|/K \rceil = O(\log n)$ input tokens from the gathered representation and emits them as the depth-0 representation $x_i^0$ (Equation 13). This partitions the input across the $K$ experts and forms the leaves of $G(dp)$.

For each interior node $(l, i)$, we use the aggregation function $AGG_{\text{trans}}$ to realize the DP transition $f(\boldsymbol{n}, i, s_{\boldsymbol{g}(\boldsymbol{n},i)}, dp(\boldsymbol{h}(\boldsymbol{n},i)))$ in Eq. 24:

$$x_i^l \;=\; AGG_{\text{trans}}\big(\{x_j^k \mid (k,j) \in A_i^l\}\big) \;\approx\; dp(i), \tag{28}$$

where $A_i^l$ collects both the input-token predecessors $\{(0,j) : j \in \boldsymbol{g}(\boldsymbol{n},i)\}$ and the subproblem predecessors $\{(d(j),j) : j \in \boldsymbol{h}(\boldsymbol{n},i)\}$. Since $J$ and $P$ in Eq. 24 are constants (Assumption 4.3 of Feng et al. (2023)), $|A_i^l|$ is constant, so $AGG_{\text{trans}}$ acts on a constant-size multiset. We implement $AGG_{\text{trans}}$ as a multiset-injective sum (or min/max) followed by an MLP (Zaheer et al., 2017; Xu et al., 2019); by Assumption 4.3 of Feng et al. (2023), $f$ can be approximated by such an MLP of constant size with parameter values bounded by $O(\text{poly}(n))$. Since the same module is reused at every depth, the per-iteration parameter count is $O(\text{poly}(K))$ and is independent of the DAG depth $L(dp)$.

At the final depth, the readout (Equation 15) implements the DP aggregation $u(\square_{i \in \mathcal{A}_{\boldsymbol{n}}} dp(i))$:

$$y \;=\; AGG_{\text{out}}\Big(\{x_j^k \mid (k,j) \in A_1^{L(dp)}\}\Big) \;\approx\; u(\square_{i \in \mathcal{A}_{\boldsymbol{n}}} dp(i)), \tag{29}$$

where $AGG_{\text{out}}$ is taken as $\square$ (one of $\sum, \min, \max$) followed by $u$, both realized by a constant-size MLP per Assumptions 4.3 and 4.5 of Feng et al. (2023). Composing Blocks 1 and 2 thus yields a Transformer of constant attention depth, with $L(dp)$ DAG-MoE iterations sharing parameters, hidden dimension $O(K \log n)$, and parameter values bounded by $O(\text{poly}(n))$, that outputs the correct DP answer at the $[\text{ANS}]$ position. $\qquad \square$

## B. More details on the model implementation and training

In this section, we provide more details on the model implementation and training. The code for reproducibility is available at the anonymous link https://anonymous.4open.science/r/DAG_MoE-1301/.

## B.1. Pretraining

*Table 6.* The model configuration and hyper-parameter setting for DAG-MoE and baseline

| configuration | DAG-MoE-s & MoE-s | DAG-MoE-m & MoE-m | DAG-MoE-l & MoE-l |
|---|---|---|---|
| model hidden size $d$ | 512 | 512 | 1024 |
| number of layer | 4 | 6 | 8 |
| number of attention heads | 32 | 32 | 32 |
| number of key-value heads | 8 | 8 | 8 |
| number of experts $N$ | 32/64 | 32 | 32 |
| expert hidden size $d_r$ | 256/128 | 256 | 512 |
| balance loss coefficient | 0.01 | 0.01 | 0.01 |
| Router Z loss coefficient | 0.001 | 0.001 | 0.001 |
| dropout ratio | 0.0 | 0.0 | 0.0 |
| optimizer | adamW | adamW | adamW |
| adam $\beta_1$ | 0.9 | 0.9 | 0.9 |
| adam $\beta_2$ | 0.999 | 0.999 | 0.999 |
| adam $\epsilon$ | 1e-8 | 1e-8 | 1e-8 |
| weight decay | 0.1 | 0.1 | 0.1 |
| learning rate | 5e-4 | 5e-4 | 3e-4 |
| warmup steps | 2000 | 2000 | 2000 |
| decay ratios | 0.2 | 0.2 | 0.2 |
| learning rate scheduler | WSD | WSD | WSD |
| batch size | 256 | 256 | 512 |

We summarize the configuration and hyperparameters of DAG-MoE and the baseline in Table 6 and elaborate below.

**Model configuration.** For both the baseline MoE and DAG-MoE, we build on top of Llama3.1-8B (Dubey et al., 2024). We retain its tokenizer, vocabulary, attention module, and FFN design, but reduce the number of Transformer layers and the hidden dimension due to resource constraints. The MoE module follows the standard token-choice router from Switch Transformer (Fedus et al., 2022) with a balance loss, except that expert scores are computed with a Sigmoid function instead of SoftMax, which we find performs better in practice. In addition, we apply a router Z-loss to regularize the logits, following recent state-of-the-art MoE models (Muennighoff et al., 2025; Zoph et al., 2022; Tian et al., 2025). For DAG-MoE, we incorporate the DAG learning module on top of the standard MoE block, varying both the graph dimension $d_g$ and the depth $L$ during evaluation. For the baseline MoE, we include a shared expert to ensure parameter parity with DAG-MoE. The shared expert adopts the same architecture as other experts, and its hidden dimension $d_r$ is adjusted to match the additional parameters introduced by the DAG learning module. Our implementation is based on the Hugging Face Transformers library (Wolf et al., 2019).

**Data.** We use the Pile (Gao et al., 2020), a large-scale open-source pretraining corpus, and conduct two sets of pretraining. For DAG-MoE-s and DAG-MoE-m, we randomly sample a subset of 10 million Pile documents and reserve 10% as the evaluation set, resulting in about 12B tokens for training and 1.3B for evaluation. For DAG-MoE-l, we use data streaming to train the model on the Pile for 37,500 steps, resulting in about 40B training tokens. Detailed data statistics can be found in Table 7. We split the original documents into sub-samples of sequence length 2048. We additionally include out-of-domain corpora for evaluating DAG-MoE-l: FineWeb-Edu (Lozhkov et al., 2024), Wikipedia text (Thrush et al., 2022), and C4 (Raffel et al., 2020), randomly sampling 500,000 documents from each as the evaluation corpus.

**Training.** We train all models from scratch on the pretraining dataset. Optimization is performed using AdamW with default settings, and the maximum learning rate is adjusted according to model size. Following prior work (Bae et al., 2025; Hu et al., 2024; Tian et al., 2025), we employ the WSD (warmup–stable–decay) scheduler, which not only improves convergence but also enables checkpoint reuse during training. The warmup phase is fixed to 2,000 steps, and the decay

*Table 7.* Pretraining data statistics. The 12B subset is randomly sampled with a fixed train/val split; the 40B run uses streaming, so per-sample counts and a held-out validation split do not apply.

| Pile | # samples | # train tokens | # val. tokens |
|---|---|---|---|
| 12B | 10,000,000 | $1.25 \times 10^{10}$ | $1.39 \times 10^9$ |
| 40B (streaming) | — | $3.93 \times 10^{10}$ | — |

ratio is set to 20%. Training is conducted on the causal language modeling task with cross-entropy loss. The coefficients for the balance loss and Z-loss are set to 0.01 and 0.001, respectively. We apply a weight decay of 0.1, and the global batch size is varied across models. All experiments are run with DeepSpeed ZeRO-2 (Rajbhandari et al., 2020) on 8 NVIDIA A100 GPUs. Our implementation builds on LlamaFactory (Zheng et al., 2024).

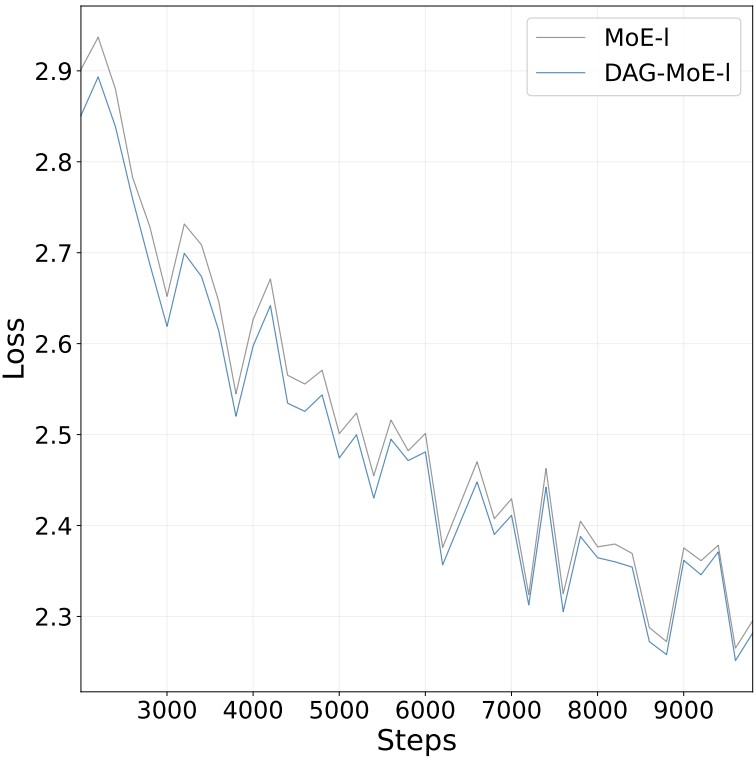

*Figure 6.* Pretraining loss curves of DAG-MoE-l and MoE-l.

### B.2. Fine-tuning and downstream evaluation

**Model configuration.** For fine-tuning, we directly use the pretrained DAG-MoE-l and MoE-l as the base models. For DAG-MoE-l, we set $d_g = 256$ and $L = 2$ in the DAG learning module. Correspondingly, for MoE-l, we add a shared expert with hidden size 512, so that both models have 699M parameters. See Table 6 for the detailed model configuration.

**Data.** For fine-tuning, we use a mixture of Alpaca (Taori et al., 2023), Open-Platypus (Lee et al., 2023), SlimOrca (Mukherjee et al., 2023), MathInstruct (Yue et al., 2023), Open-R1-Math[2], and MetaMathQA (Yu et al., 2023), without any up- or down-sampling. All datasets are obtained from Hugging Face.

**Training.** We train the model for 3 epochs on all the data with a constant learning rate of $2\mathrm{e}{-}5$. The coefficients for balance loss and Z-loss are the same as in pretraining, while we set the weight decay to 0 to allow the model to adjust its behavior under fine-tuning. We use a batch size of 256, which results in 15,435 total steps for 3 epochs. All experiments are run with DeepSpeed ZeRO-2 (Rajbhandari et al., 2020) on 8 NVIDIA A100 GPUs. Our implementation builds on LlamaFactory (Zheng et al., 2024).

**Evaluation.** After fine-tuning, we evaluate both DAG-MoE-l and MoE-l on downstream tasks including PIQA (Bisk et al., 2020), ARC-e (Clark et al., 2018), HellaSwag (Zellers et al., 2019), GPQA (Rein et al., 2024), Lambada (Paperno et al., 2016), MMLU (Hendrycks et al., 2021), and BBH (Suzgun et al., 2023). PIQA tests physical commonsense by asking the model to choose the more plausible solution to an everyday physical task. ARC-e is the "Easy" subset of the AI2 Reasoning Challenge, consisting of grade-school science multiple-choice questions. HellaSwag evaluates grounded commonsense by requiring the model to select a plausible continuation for a short scenario. GPQA measures expert-level knowledge and reasoning with graduate-level multiple-choice questions. Lambada assesses broad-context language modeling via

---

[2]https://huggingface.co/datasets/open-r1/OpenR1-Math-220k

last-word prediction that requires understanding a long passage. MMLU benchmarks multi-task language understanding across 57 academic subjects in a few-shot multiple-choice format. BBH (BIG-Bench Hard) comprises 23 challenging reasoning tasks that probe compositionality, logic, and algorithmic generalization. We use a 10-shot setting for HellaSwag and a 5-shot setting for MMLU; all other benchmarks use a 0-shot setting without CoT. Evaluation is conducted through OpenCompass (Contributors, 2023).

## C. Additional discussion on the experiments

### C.1. Pretraining of DAG-MoE-l

We present the pretraining loss curves of DAG-MoE-l and MoE-l from 1,000 to 10,000 steps in Fig. 6. DAG-MoE-l exhibits substantially faster early-stage convergence, with a clear loss gap that supports the claim that DAG-MoE offers greater flexibility than standard MoE. As training progresses, the gap narrows and stabilizes toward the end. We hypothesize that this is because both models are relatively small and reach similar optima at this data scale. We plan to evaluate DAG-MoE at larger model sizes and on larger training corpora in future work.

### C.2. Visualization of the learned DAG structure

To inspect what DAG-MoE actually learns inside its DAG learning module, we analyze a DAG-MoE-s model (4 layers, 32 experts, top-$K{=}4$, $L{=}2$) trained on 12B tokens from the Pile, evaluated on a held-out subset. Although the implementation in Eq. 9–Eq. 10 does not produce explicit scalar edge weights, we use the per-edge contribution $\hat{x}^l_{(i,j)}$ in Eq. 10 as a natural proxy: its $\ell_2$ norm $\|\hat{x}^l_{(i,j)}\|_2$ measures how strongly source node $j$ influences target node $i$ at depth $l$, and we treat it as the edge weight from expert $j$ to expert $i$ at iteration $l$.

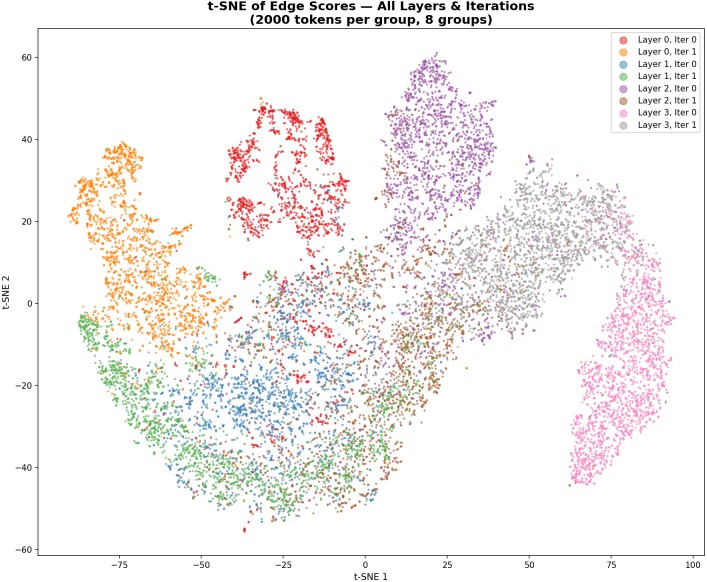

*Figure 7.* t-SNE projection of the flattened $K \times K$ edge-weight vector for each token, colored by the (layer, iteration) pair.

**Per-token structural patterns.** To examine how the learned structure varies across tokens, we flatten the $K \times K$ edge-weight matrix at each (layer, iteration) into a single vector and project all per-token vectors with t-SNE. As shown in Fig. 7, points cluster sharply by their (layer, iteration) label, confirming that different stages learn distinct structural patterns. Within each cluster, the per-token vectors still spread over a noticeable region, indicating that the learned structure is also *input-adaptive* rather than a single fixed graph reused across all tokens.

**Mean edge weights across layers.** Fig. 8 shows the mean edge-weight matrix at each (layer, iteration) pair, averaged over a randomly sampled held-out batch. Across both layers and iterations, source experts exhibit clear preferences for specific target experts rather than producing uniform aggregation, indicating that the learned DAG implements a non-trivial, non-symmetric interaction structure rather than collapsing to a permutation-invariant sum. Together with the t-SNE visualization

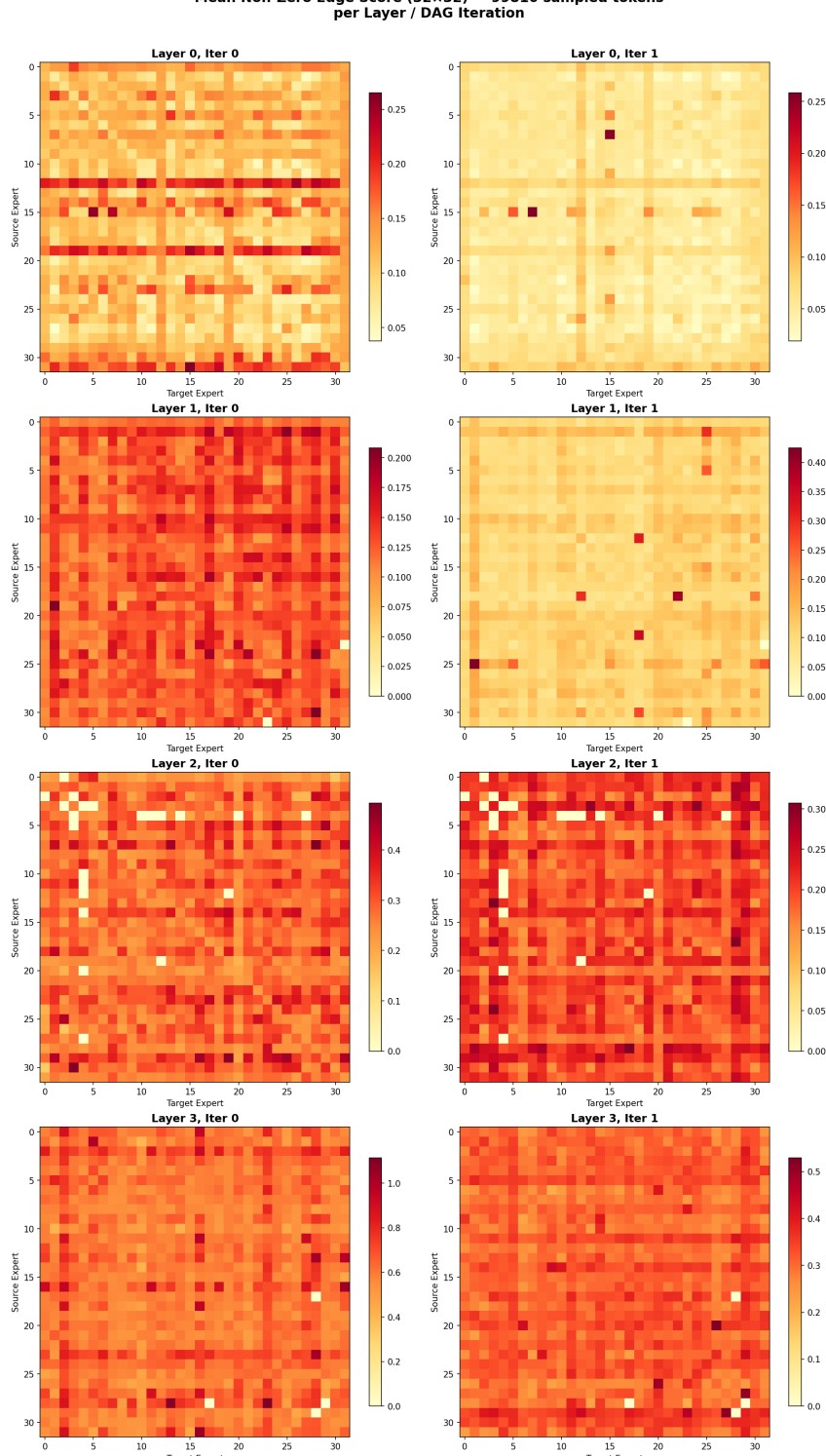

*Figure 8.* Mean edge-weight heatmaps across layers (rows) and DAG iterations (columns) of DAG-MoE-s. Each cell shows the mean of $\|\hat{x}^l_{(i,j)}\|_2$ over a held-out batch, with rows of each heatmap indexing target experts and columns indexing source experts.

above, these results support the picture that DAG-MoE discovers diverse, layer-specific, and token-dependent aggregation structures during training.

