# OpenReview forum: "DAG-MoE: From Simple Mixture to Structural Aggregation in Mixture-of-Experts"
_ICML.cc/2026/Conference — ICML 2026 regular_

### Official Review · Reviewer_7AFr · 2026-02-14

**Soundness:** 4
**Presentation:** 2
**Significance:** 3
**Originality:** 3
**Overall Recommendation:** 4
**Confidence:** 4

**Summary:**

This paper proposes DAG-MoE, a interesting MoE architecture that replaces permutation-invariant weighted summation with DAG-based structural aggregation of expert outputs. By learning a directed acyclic graph among selected experts, DAG-MoE substantially expands the expert-combination space without modifying experts or routers, and enables multi-step reasoning within a single MoE layer. Both theoretical analysis and extensive experiments show that DAG-MoE achieves consistently better performance than standard MoE in pretraining and downstream tasks.

**Compliance With Llm Reviewing Policy:**

Affirmed.

**Final Justification:**

I look forward to more substantial theoretical analysis in the main body of the paper.

**Key Questions For Authors:**

See Weaknesses.

**Limitations:**

yes

**Strengths And Weaknesses:**

- Strengths
1. `Structural aggregation beyond weighted sums`. The paper introduces DAG-based expert aggregation, which significantly expands the expert-combination space without changing expert or router configurations, overcoming the expressiveness limits of standard MoE weighted summation.
2. `Improved reasoning capability within a single MoE layer`. By organizing experts as a directed acyclic graph, the model can approximate multi-step reasoning ($e.g.$, dynamic programming) inside one MoE layer, effectively increasing logical depth without additional layers.
3. `Consistent empirical gains with modest overhead`. Extensive pretraining and fine-tuning experiments show consistent performance improvements over standard MoE under similar parameter and compute budgets, demonstrating strong practicality and scalability.
4. `Open-sourcing`. The authors provide anonymized open-source code in the appendix, which is highly commendable. However, due to the long training time and limited reviewer availability, a thorough evaluation is not currently feasible. Overall, the open-source spirit deserves praise. Additionally, it is recommended that the code link be included in the main text to better attract readers' attention and engagement.

- Weaknesses
1. Although the proposed DAG-MoE appears promising, the additional computational overhead compared to standard MoE methods is non-negligible. Moreover, the combination of MoE with DAG structures has been explored in prior work[1].
2. Many key proofs are deferred to the appendix, which significantly affects readability. For example, crucial parts such as the proof that DAG-style MoE is more expressive than standard MoE and its reasoning advantages are postponed to the appendix. Moreover, the logical connections between different theoretical results are not sufficiently clear and appear somewhat fragmented.
3. The authors theoretically argue that the proposed method is well-suited for solving DP problems. However, DP is inherently complex, and its corresponding DAG structures are often fixed. The proposed method only outlines a vague direction regarding DAG construction. It remains unclear whether the adaptively learned DAG can align with the DAG structure required by DP problems. Additionally, a clearer mapping between the black-box model parameters and the fixed solution structure of DP problems is needed.
4. The evaluation focuses solely on standard MoE and overlooks other MoE variants [2-3]. The authors are encouraged to incorporate more recent work to provide a more comprehensive validation. Furthermore, it is recommended to include additional ablation studies on small-scale tasks.

Other: The related work section is placed after the methodology section, which is uncommon. It is recommended to position it before the methodology (preliminaries) section or before the conclusion section.

[1] DiEP: Adaptive Mixture-of-Experts Compression through Differentiable Expert Pruning
[2] HydraLoRA: An Asymmetric LoRA Architecture for Efficient Fine-Tuning
[3] CoLA: Collaborative Low-Rank Adaptation

---

> ### Author Rebuttal · Authors · 2026-03-31
>
> We sincerely appreciate the valuable feedback from the reviewer.
>
> **W1**: additional computation overhead.
>
> **AW1**: We would like to clarify that the FLOPs of DAG-MoE are comparable to standard MoE, as analyzed in Section 3.3. Furthermore, in all empirical experiments, we control the total parameter count of the standard MoE baseline to match DAG-MoE, ensuring a fair comparison. We additionally provide wall-clock training throughput measurements below:
>
> | Model               | Total Params (ratio) | Total FLOPs (E) (ratio)  | # tokens/s (M) (ratio) |
> | ------------------- | ------------ | --------------- | ---------- |
> | Standard MoE , topk=4   | 184,750,592 (100%) | 8.931 (100%)           | 0.7744 (100%)    |
> | DAG-MoE-s, topk=4 , L=1 | 184,687,104 (99.97%) | 8.926 (99.94%)          | 0.7627 (98.49%)     |
> | DAG-MoE-s, topk=4 , L=2 | 185,016,832 (100.14%)  | 8.951 (100.22%)          | 0.7404  (95.61%)   |
>
> We can see that DAG-MoE didn’t introduce a huge overhead to the baseline MoE.  Finally,  DAG-MoE is more efficient than prior structured alternatives: CoE requires an independent router at each selection stage, adding routing cost proportional to the number of iterations. DAG-MoE routes only once and introduces structural aggregation as a lightweight add-on, keeping routing cost identical to standard MoE.
>
> Thanks for bringing DiEP, which is an interesting work. DiEP models an MoE layer as a two-node DAG (input → output) solely as a formal framework for defining a differentiable expert pruning search space. DAG-MoE, in contrast, introduces a multi-depth DAG over the outputs of selected experts to learn structural aggregation relationships. The two works target different problems — model compression vs. model architecture. We will add a clarifying discussion in the related work section.
>
> **W2**: Proof in appendix.
>
> **AW2**: Due to ICML's page limit, detailed proofs are deferred to the appendix, which is standard practice. The three results form a coherent chain: Proposition 3.1 establishes that DAG-style MoE can injectively encode any DAG structure; Theorem 3.2 uses this to prove strict expressiveness superiority over standard MoE; and Theorem 3.3 shows this expressiveness enables multi-step reasoning within a single layer — providing the practical rationale for replacing weighted summation with structural aggregation. We will include this explanation at the beginning of Section 3.2 in the revision.
>
> **W3**: gap between DP theorem and implementation.
>
> **AW3**: Theorem 3.3 is a capacity result, not a learnability claim: it establishes that a DAG-MoE configuration exists that can simulate DP solving — analogous to universal approximation theorems, which prove representational capacity without guaranteeing gradient descent finds the target. We acknowledge that the gap between "capacity exists" and "gradient descent learns the aligned structure" is a genuine theoretical limitation, and we will make this distinction explicit in the revision.
>
> Regarding alignment with fixed DP structures: the paper does not claim DAG-MoE literally learns to solve DP problems. The DP discussion is a theoretical motivation showing that the structural form of DAG aggregation naturally fits multi-step compositional computation. The adaptively learned DAG does not need to exactly match a specific DP graph — it needs only to capture the spirit of hierarchical, ordered aggregation to benefit reasoning tasks. Empirical support comes from downstream results: GPQA (+6.06), requiring multi-step graduate-level reasoning, shows the largest gain, while pattern-matching benchmarks show near-zero change.
>
> **W4**: Additional comparison.
>
> **AW4**:  We thank the reviewers for constructive suggestions.  Both HydraLoRA and CoLA are parameter-efficient adapters added on top of a pre-trained backbone. We focus on the pre-trained backbone arch itself. Both the training tasks and model arch settings are very different. To this end, we provide an additional empirical comparison against Chain-of-Experts (CoE) [1], which also introduces iterative, multi-stage expert processing. We reimplement CoE under our codebase and add a shared expert to all baselines to ensure fair parameter matching. Results on DAG-MoE-s (top-K=4, 12B tokens) are as follows:
>
> | Model                                 | Add. Params/Layer | PPL improvement  |
> | ------------------------------------- | ----------------- | ------ |
> | Standard MoE     | 0                 | 0 |
> | Standard MoE + shared expert w/ d_g=256| 393,216           | 0.433 |
> | CoE, L=2, d_g=256                 | 393,216           | 0.480 |
> | DAG-MoE, L=2, d_g=128                  | 393,216           | 0.587 |
>
> DAG-MoE outperforms both standard MoE and CoE under equal or fewer added parameters.
>
> **W5**: Placement of the related work section.
>
> **AW5**: We thank the reviewer for the suggestion. We will move the related work section to appear before the methodology section in the revision.

---

> > ### Author Rebuttal · Reviewer_7AFr · 2026-04-02
> >
> > I look forward to more substantial theoretical analysis in the main body of the paper.

---

> > > ### Author Response · Authors · 2026-04-03
> > >
> > > Thank you again for all your insightful suggestions! In the revision, we will restructure the theory section at main paper to improve its flow: we will open with an explicit roadmap showing how the three results build on each other — Proposition 3.1 (structural encoding) → Theorem 3.2 (expressiveness) → Theorem 3.3 (reasoning benefit) — and add proof sketches for Theorem 3.2 and 3.3 in the main body so readers can follow the logical progression without needing to consult the appendix.

---

### Official Review · Reviewer_DzVj · 2026-03-10

**Soundness:** 3
**Presentation:** 2
**Significance:** 2
**Originality:** 3
**Overall Recommendation:** 4
**Confidence:** 4

**Summary:**

This paper investigates the aggregation problem in the MoE model. Unlike previous papers that focused on improving routers or expert granularity, this paper shifts its attention to a long-standing default approach—the aggregation of expert outputs. The paper proposes replacing the simple weighted summation in standard MoE with a learnable DAG-based aggregation structure. This expands the expert-combination space without changing the configuration of experts and routers, and may even approximate multi-step reasoning in a single layer.

**Compliance With Llm Reviewing Policy:**

Affirmed.

**Final Justification:**

I think most of the concerns have been well solved, so I maintain my positive score.

**Key Questions For Authors:**

Please see the "Weaknesses" above.

**Limitations:**

yes

**Strengths And Weaknesses:**

Strengths:

+ Different from existing MoE-based methods that focus more on how to select experts or how to design routers and achieve load balance,  this paper takes a different approach, exploring "how to combine the results of selected experts," which is a relatively novel perspective.

+ The proposed DAG-based aggregation method is reasonable and is compatible with the existing MoE framework.

+ Experimental results on different tasks demonstrate the efficiency of the proposed method. Besides that, the authors' claim is also marginally verified by the theoretical analysis.

Weaknesses:

+ The paper presents many strong conclusions, such as enhanced expressive power and effects similar to multi-step reasoning. However, the experiments primarily focus on perplexity and general benchmarks, and cannot directly prove these deeper conclusions.

+ Current comparisons primarily focus on methods like standard MoE and shared expert. However, there hasn't been a sufficient comparison of more direct alternatives, such as: 1) Adding an attention mixer to the expert output; 2) Adding a small MLP mixing module, etc. Therefore, it's not yet entirely certain whether the improvement is due to the "DAG structure" itself or because of "adding a more complex mixing layer."

+ It is suggested to add more visualization results to demonstrate the efficiency of DAG-based aggregation method, i.e., visualize the learned edge weights, etc.

---

> ### Author Rebuttal · Authors · 2026-03-31
>
> We sincerely appreciate the positive feedback and the valuable suggestions!
>
> **W1**: theoretical and empirical mismatch.
>
> AW1: Thanks for the insightful question. Perplexity directly measures how well the model captures the distribution of natural language, which inherently reflects expressive capacity. We evaluate DAG-MoE across diverse configurations and data scales, consistently demonstrating lower perplexity than standard MoE. Theorem 3.2 establishes that this greater expressive capacity exists in the architecture; the empirical results confirm that it is realized during training. Regarding multi-step reasoning, DAG-MoE yields its largest gains on GPQA (+6.06) and PIQA (+3.15), which require multi-step inference, while HellaSwag shows near-zero change — consistent with the reasoning benefit.
>
> **W2**: comparison to other mixing types.
>
> Here, we provide additional experiments on changing the DAG module to a MLP mixing module. Specifically, we concatenate all expert embeddings and use an MLP to process the concatenated vector to get the final output. We adjust the hidden size in MLP to match the parameter count. Here are the results:
>
> | Model                                | Add. Params/Layer | Eval Loss ↓ |
> | ------------------------------------ | ----------------- | ----------- |
> | Standard MoE      | 0                 | 2.7168      |
> | Standard MoE + shared expert w/ d_g=32 | 49,152            | 2.7072      |
> | Standard MoE + shared expert w/ d_g=64 | 98,304            | 2.7012      |
> | MLP mixing, d_g=32                   | 49,152            | 2.8406      |
> | MLP mixing, d_g=64                   | 98,304            | 2.8006      |
> | DAG-MoE-s, d_g=32, L=1               | 36,864            | 2.6948      |
> | DAG-MoE-s, d_g=32, L=2               | 73,728            | 2.6899      |
>
> We can see that natively adding MLP mixing instead hurts performance.
>
> **W3**: More visualization results.
>
> **AW3**:  We thank the reviewer for this suggestion. Although there are no explicit scalar edge weights in our implementation, we propose a proxy measure to analyze the learned aggregation structure. Specifically, we select a DAG-MoE model (4 layers, 32 expert top-K=4, L=2) trained on 12B tokens, and on an unseen held-out dataset, we compute the L2 norm of $\hat{x}^l_{(i,j)}$ (Eq. 10) for each token, which captures how strongly expert $i$ influences expert $j$ at depth $l$ and we utilize this as the edge weight. We plot the mean edge weight scores across randomly sampled tokens at each layer and DAG iteration. The heatmaps (https://anonymous.4open.science/r/DAG_MoE-1301/edge_scores_NxN_mean.png) show that at different layers and iterations, source experts exhibit clear preferences for specific target experts, demonstrating a diverse and non-trivial interaction structure.  Finally, we further flatten the K×K edge weights per token into a single vector and apply t-SNE to visualize the distribution across layers and iterations. The resulting plot (https://anonymous.4open.science/r/DAG_MoE-1301/edge_scores_tsne_combined.png) shows that different layers and iterations form distinct clusters, confirming that different structural patterns are learned at each stage. Within each cluster, the edge scores span a wide distribution, indicating that the learned structure is also input-dependent rather than fixed. Together, these results demonstrate that DAG-MoE learns diverse and adaptive structural patterns during training.

---

> > ### Author Rebuttal · Reviewer_DzVj · 2026-04-03
> >
> > I maintain my positive score.

---

> > > ### Author Response · Authors · 2026-04-07
> > >
> > > We sincerely thank the reviewer for the positive feedback and thoughtful suggestions. We will incorporate all of this feedback into the future version.

---

### Official Review · Reviewer_5fGo · 2026-03-12

**Soundness:** 3
**Presentation:** 2
**Significance:** 2
**Originality:** 3
**Overall Recommendation:** 4
**Confidence:** 3

**Summary:**

The paper proposes DAG-MoE, an innovative MoE architecture that explores expert-output mixture as a complementary scaling axis. DAG-MoE introduces a framework to learn the optimal aggregation structure among the selected experts. The authors provide analysis on both performance and efficiency, and the experimental results demonstrate that DAG-MoE is effective while not introducing additional FLOPs compared to standard MoE architectures.

**Compliance With Llm Reviewing Policy:**

Affirmed.

**Final Justification:**

My final recommendation is **weak accept**. The rebuttal addressed my main concerns and I will **raise my score to 4**.

**Key Questions For Authors:**

See weaknesses.

**Limitations:**

yes

**Strengths And Weaknesses:**

**Strengths**

1. The architecture design of DAG-MoE is innovative and interesting. It is a meaningful exploration of alternative MoE architecture. DAG-MoE leverages expert representations from a fresh perspective.

2. The pipeline of DAG-MoE is described clearly and in detail, making the paper easy to follow.

**Weaknesses**

1. Despite the analysis of computational cost suggests that DAG-MoE does not introduce more FLOPs than adding an additional expert, the wall-clock time remains a concern. In practical deployment, wall-clock time is a crucial metric of efficiency, could the authors provide the comparison of wall-clock time between DAG-MoE and vanilla MoE?
2. The concerns about fair comparison in experiments: Does DAG-MoE use the same expert setting (e.g., the number of experts and parameter size of each expert) as vanilla MoE ? If so, does DAG-MoE introduce additional parameters because several parameters are introduced ($W_{down}, W_{edge}, W_{node}$)?
3. Lack of sufficient ablation study: The hyperparameter $L$ (the number of iterations) appears to be important in DAG-MoE, but there is no ablation study is provided, could the authors provide ablation study on $L$?
4. As a work focusing on MoE architecture design, the paper lacks comparison with other specialized MoE architecture (e.g., Chain-of-Experts).

---

> ### Author Rebuttal · Authors · 2026-03-31
>
> We sincerely appreciate the valuable feedback from the reviewer.
>
> **W1**: wall-clock time comparison.
>
> **AW1**: Thanks for the valuable suggestion. We provide a wall-clock training throughput comparison below:
>
> | Model               | Total Params (ratio) | Total FLOPs (E) (ratio)  | # tokens/s (M) (ratio) |
> | ------------------- | ------------ | --------------- | ---------- |
> | Standard MoE , topk=4   | 184,750,592 (100%) | 8.931 (100%)           | 0.7744 (100%)    |
> | DAG-MoE-s, topk=4 , L=1 | 184,687,104 (99.97%) | 8.926 (99.94%)          | 0.7627 (98.49%)     |
> | DAG-MoE-s, topk=4 , L=2 | 185,016,832 (100.14%)  | 8.951 (100.22%)          | 0.7404  (95.61%)   |
>
> We can see that DAG-MoE introduces only 1.51% overhead at L=1 and 4.49% at L=2 relative to standard MoE — a modest cost for the consistent perplexity gains demonstrated in the paper. Additionally, system-level optimizations, such as CUDA kernel fusion (torch.complie), could further reduce this overhead, though we consider such engineering work outside the scope of this paper. Finally, the primary focus of this paper is to introduce DAG-based structural aggregation as a new framework for expert mixture in MoE. The current implementation is one concrete instantiation; we believe there exist alternative implementations that could achieve better efficiency and performance, and we leave this exploration to future work.
>
> **W2**: The concerns about fair comparison in experiments:
>
> **AW2**: Yes, we ensure fair comparison by enforcing DAG-MoE and the baselines to have the matching parameter count and the identical expert configuration. The DAG learning module does introduce additional parameters. To ensure fair comparison, we add a shared expert to the baseline MoE, where the shared expert's parameter count is set to match the additional parameters introduced by the DAG learning module. This is reflected in the "Total add parameter" in Figure 3.  DAG-MoE outperforms the baseline MoE+shared-expert with matching parameters.
>
>
> **W3**: Lack of sufficient ablation study.
>
> **AW3**: We respectfully point out that ablation on the number of iterations L is already provided in the paper. Figure 3 presents perplexity curves for L ∈ {1, 2, 3} across two model sizes and two expert granularities, and Figure 4 directly plots the perplexity improvement of each L value over the standard MoE baseline. Key findings are: (1) increasing L from 1 to 2 yields substantial gains, (2) L=3 provides only marginal additional improvement, suggesting L=2 is a favorable operating point.
>
> **W4**: As a work focusing on MoE architecture design, the paper lacks comparison with other specialized MoE architecture (e.g., Chain-of-Experts
>
> **AW4**: Here we provide an empirical comparison with Chain-of-Experts (CoE) [1], which is the most closely related specialized MoE architecture involving iterative multi-stage expert processing. We reimplement CoE under our codebase with matched parameters and train on the same 12B token dataset:
>
> | Model                                 | Add. Params/Layer | PPL improvement|
> | ------------------------------------- | ----------------- | ------ |
> | Standard MoE     | 0                 | 0 |
> | Standard MoE + shared expert w/ d_g=256 | 393,216           | 0.433 |
> | CoE, L=2, d_g=256                 | 393,216           | 0.480 |
> | DAG-MoE, L=2, d_g=128                  | 393,216           | **0.587** |
>
>
> DAG-MoE outperforms CoE under equal or fewer parameters in both settings.
>
> [1] Zihan Wang, et al., Chain-of-Experts: Unlocking the Communication Power of Mixture-of-Experts Models, arxiv, 2025

---

> > ### Author Rebuttal · Reviewer_5fGo · 2026-04-02
> >
> > Thanks for the detailed explanations in such a short time. DAG-MoE is a meaningful exploration of MoE architecture and the rebuttal response address my concerns.
> >
> > I will **raise my score: 3 -> 4**.

---

> > > ### Author Response · Authors · 2026-04-03
> > >
> > > We sincerely thank the reviewer for all the constructive feedback! We will ensure that all aspects are incorporated in the revision.

---

### Official Review · Reviewer_zUzT · 2026-03-14

**Soundness:** 2
**Presentation:** 3
**Significance:** 3
**Originality:** 4
**Overall Recommendation:** 4
**Confidence:** 4

**Summary:**

The paper introduces DAG learning module as an additional component in the MoE architecture. The paper then addresses the question how the selected expert outputs are aggregated after routing. Instead of the standard permutation-invariant weighted sum, the authors propose to aggregate the top-K expert outputs through a learned directed acyclic graph, yielding the proposed DAG-MoE architecture. The article's central contribution is the claim that structural aggregation can expand the effective composition space of selected experts without changing the expert bank or router, and may also enable limited multi-step computation

**Compliance With Llm Reviewing Policy:**

Affirmed.

**Key Questions For Authors:**

How does DAG dependencies affect gradient propagation, training stability and routing convergence?

Are there any cases where the DAG component would be more suitable than others? Can you verify it empirically?

The theory is stated for a general DAG-style MoE with injective aggregation, but the implemented DAG-learning module uses a much more restricted structure: fixed K nodes per iteration, only depth-(l−1) to depth-l interactions, residual carry-over, and final summation. Which theoretical claims still provably apply to the actual instantiated architecture, and which do not?

**Limitations:**

The authors do mention some technical limitations, including restrictions on the DAG class and the small scale of evaluation.

**Strengths And Weaknesses:**

Strengths:

The observation that standard MoE aggregation is permutation-invariant and therefore structurally limited is reasonable, and the proposal to move from simple mixing to learned structural composition is interesting and relevant.

 it's a valid question to ask if there is a more effective way other than summation to combine information from the experts?

The method is modular: it can be inserted on top of a standard sparse MoE block without redesigning the router itself

The empirical section shows consistent perplexity improvements over the authors’ baseline

The concepts are well explained

Weaknesses:


the experiments do not test dynamic programming tasks, algorithmic reasoning, or any controlled benchmark that would directly validate the “multi-step reasoning within a single MoE layer” claim. In its current form, that part reads as speculative and insufficiently connected to the empirical section.

The experiments are on a fairly small scale.

The comparison is mainly against a standard MoE plus an added shared expert to roughly match parameter count. That is a reasonable sanity baseline, but it is not sufficient for the paper’s broader structural claims.

The related-work section discusses iterative or structural variants such as S’MoRE and Chain-of-Experts-like ideas, yet there is no empirical comparison against any structurally richer or iterative aggregation baseline.
it is hard to know whether the gains come specifically from DAG structure, from adding another nonlinear mixing stage, or simply from increasing depth/capacity in the MoE block in a different form.

---

> ### Author Rebuttal · Authors · 2026-03-31
>
> We sincerely appreciate the valuable feedback from the reviewer.
>
> **W1**: Regarding dynamic programming and multi-step reasoning.
>
> **AW1**: We acknowledge this concern. Theorem 3.3 is a capacity result, not a claim that training will reproduce DP solving behavior — analogous to universal approximation theorems, which prove representational capacity without guaranteeing gradient descent finds the target. The DP discussion is a theoretical motivation for why structural aggregation benefits compositional tasks. Empirically, DAG-MoE yields its largest gains on GPQA (+6.06) and PIQA (+3.15), which require multi-step inference, while HellaSwag shows near-zero change — consistent with the reasoning benefit.
>
> **W2**: The experiments are on a fairly small scale.
>
> **AW2**: We acknowledge that, due to computational budget limitations, we were unable to train models at the billion-parameter scale, and we plan to explore this in future work. That said, we would like to clarify that our claim is not state-of-the-art performance at scale, but that structural aggregation consistently improves over standard MoE — validated across: three model sizes (12B and 40B token regimes), two expert granularities (top-4 and top-8), multiple hyperparameter settings (d_g ∈ {64,128}, L ∈ {1,2,3}), and four evaluation corpora including three out-of-domain datasets. DAG-MoE outperforms the parameter-matched baseline across all 20+ configurations, following the established validation practice of prior MoE architectural works [1, 2].
>
> **W3&W4**: Insufficient comparison with the advanced MoE baseline.
> AW3:  Here we provide an additional empirical comparison against Chain-of-Experts (CoE) [3], which also introduces iterative, multi-stage expert processing. We reimplemented CoE under our codebase and ensured matching parameters and computation budget for the fair comparison. Results on DAG-MoE-s (top-K=4, 12B tokens) are as follows:
>
> | Model                                 | Add. Params/Layer | PPL improvement  |
> | ------------------------------------- | ----------------- | ------ |
> | Standard MoE     | 0                 | 0 |
> | Standard MoE + shared expert w/ d_g=256 | 393,216           | 0.433 |
> | CoE, L=2, d_g=256                 | 393,216           | 0.480 |
> | DAG-MoE, L=2, d_g=128                  | 393,216           | **0.587** |
>
> DAG-MoE outperforms both standard MoE and CoE under equal or fewer added parameters. We discuss S'MoRE in the related work because it is the most structurally related prior work, but direct comparison is not straightforward: it is designed as a PEFT adapter rather than a pretraining backbone, and its hierarchical expert design would require substantial modification to fit a standard token-choice setup. We leave this to future work.
>
>
> **Q1**: Impact of DAG module.
>
> **AQ1**:
> Gradient flow is maintained via residual connections at every depth (Eq. 11) and LayerNorm before each projection (Eq. 7). Training stability is ensured by zero-initializing $W^l_{up}$: at initialization, the DAG module contributes zero, reducing the block to standard MoE and allowing the model to start from a stable regime before gradually learning the DAG structure. Since the router is unchanged from standard MoE, routing convergence is unaffected. These design choices are confirmed empirically by the smooth loss curves in Fig. 6.
>
>
> **Q2**: When is the DAG component more suitable?
>
> **AQ2**:  From the downstream results, we observe that DAG-MoE achieves its largest gain on GPQA (+6.06), a graduate-level multi-step reasoning benchmark, while showing no improvement on HellaSwag and MMLU, which rely on pattern matching and knowledge retrieval. This asymmetry is consistent with Theorem 3.3: structural aggregation benefits tasks requiring compositional multi-step reasoning more than retrieval-style tasks.
>
> **Q3**: Theory and implementation mismatch.
>
> **AQ3**: We acknowledge the gap between the general DAG-style MoE formulation and our actual implementation, as noted in the Limitations section.
>
> Theorem 3.2 applies to the implementation. Even with depth-adjacent connections and fixed n(l)=K, the architecture can represent asymmetric compositions that standard MoE's permutation-invariant summation cannot — one such distinguishable pair suffices for the proof.
>
> Proposition 3.1 does not fully apply. Restricting to depth-adjacent connections and fixed K nodes per depth means only a subclass of G(K) is representable.
>
> Theorem 3.3 partially applies. Residual connections (Eq. 11) carry information across depths, partially compensating for missing skip connections, but this equivalence is not formally proven.
>
> We will make these distinctions explicit in the revision.
>
>
>
> [1] Jakub Krajewski, et al.,  Scaling Laws for Fine-Grained Mixture of Experts, arxiv, 2024.
> [2] Xu Owen He, Mixture of A Million Experts, arxiv, 2024
> [3] Zihan Wang, et al., Chain-of-Experts: Unlocking the Communication Power of Mixture-of-Experts Models, arxiv, 2025

---

> > ### Author Rebuttal · Reviewer_zUzT · 2026-04-04
> >
> > Thank you for the broad explanations and the additional experiment. It would be great to have more analysis/experiments on other nonlinear aggregation methods but I am satisfied. I maintain my score.

---

> > > ### Author Response · Authors · 2026-04-07
> > >
> > > We sincerely thank the reviewer for the constructive feedback! We will include more comparisons in a future version.

---

### Decision · Program_Chairs · 2026-04-30

**Decision:**

Accept (regular)

**Comment:**

This paper introduces DAG-MoE. This novel Mixture-of-Experts (MoE) architecture replaces the traditional permutation-invariant weighted summation of expert outputs with a learned, directed acyclic graph (DAG)-based structural aggregation. By expanding the space of expert interactions without altering the router or expert set, the framework aims to enable richer compositional behavior and potentially multi-step reasoning within a single MoE layer. Overall, reviewers agreed that the paper introduces a highly promising and original architectural direction.

Reviewers acknowledged the paper for offering a novel and meaningful perspective on MoE design. By shifting the focus from expert selection to expert aggregation, the authors address an underexplored area with a conceptually sound approach. The modular nature of DAG-MoE is a significant practical advantage, as it allows for easy integration into existing sparse MoE architectures. Furthermore, the method demonstrated consistent empirical improvements over parameter-matched baselines across pretraining and downstream tasks, which were complemented by theoretical analyses that provided useful intuition regarding increased expressiveness and structural composition.

Despite these strengths, reviewers identified several areas of concern. A primary issue is the relatively small scale of the evaluations, which raises valid questions about whether the observed gains will persist in large-scale LLM settings. Additionally, the paper's ambitious claims regarding the enablement of multi-step reasoning were not directly validated through targeted benchmarks, relying instead on indirect evidence. Reviewers also pointed out a gap between the theoretical formulation and the practical implementation, noting that the initial submission lacked comparisons against more advanced aggregation baselines. Finally, while computational FLOPs remain comparable, the method introduces a slight wall-clock overhead that warrants further analysis at scale.

The authors provided a rebuttal that successfully addressed many of these concerns. They supplied additional empirical comparisons against baselines like Chain-of-Experts and MLP mixing, which reinforced the claim that the performance gains stem specifically from structural aggregation rather than a simple increase in capacity. The rebuttal also clarified efficiency metrics, properly scoped the theoretical claims regarding capacity versus learnability, and provided further analysis of the specific scenarios—such as reasoning-heavy tasks—where DAG-MoE is most beneficial. As a result, the reviewers maintained their positive recommendations.